# SparCL: Sparse Continual Learning on the Edge

**Zifeng Wang**[1,*], **Zheng Zhan**[1,*], **Yifan Gong**[1], **Geng Yuan**[1], **Wei Niu**[2], **Tong Jian**[1],
**Bin Ren**[2], **Stratis Ioannidis**[1], **Yanzhi Wang**[1], **Jennifer Dy**[1]

[1] Northeastern University, [2] College of William and Mary

{zhan.zhe, gong.yifa, geng.yuan, yanz.wang}@northeastern.edu,
{zifengwang, jian, ioannidis, jdy}@ece.neu.edu,
wniu@email.wm.edu, bren@cs.wm.edu

## Abstract

Existing work in continual learning (CL) focuses on mitigating catastrophic forgetting, *i.e.*, model performance deterioration on past tasks when learning a new task. However, the training efficiency of a CL system is under-investigated, which limits the real-world application of CL systems under resource-limited scenarios. In this work, we propose a novel framework called Sparse Continual Learning (SparCL), which is the first study that leverages sparsity to enable cost-effective continual learning on edge devices. SparCL achieves both training acceleration and accuracy preservation through the synergy of three aspects: *weight sparsity*, *data efficiency*, and *gradient sparsity*. Specifically, we propose task-aware dynamic masking (TDM) to learn a sparse network throughout the entire CL process, dynamic data removal (DDR) to remove less informative training data, and dynamic gradient masking (DGM) to sparsify the gradient updates. Each of them not only improves efficiency, but also further mitigates catastrophic forgetting. SparCL consistently improves the training efficiency of existing state-of-the-art (SOTA) CL methods by at most $23\times$ less training FLOPs, and, surprisingly, further improves the SOTA accuracy by at most $1.7\%$. SparCL also outperforms competitive baselines obtained from adapting SOTA sparse training methods to the CL setting in both efficiency and accuracy. We also evaluate the effectiveness of SparCL on a real mobile phone, further indicating the practical potential of our method.

## 1 Introduction

The objective of Continual Learning (CL) is to enable an intelligent system to accumulate knowledge from a sequence of tasks, such that it exhibits satisfying performance on both old and new tasks [32]. Recent methods mostly focus on addressing the *catastrophic forgetting* [43] problem – learning model tends to suffer performance deterioration on previously seen tasks. However, in the real world, when CL applications are deployed in resource-limited platforms [48] such as edge devices, the learning efficiency, w.r.t. both training speed and memory footprint, are also crucial metrics of interest, yet they are rarely explored in prior CL works.

Existing CL methods can be categorized into regularization-based [2, 32, 37, 68], rehearsal-based [8, 12, 50, 61], and architecture-based [31, 42, 52, 58, 59, 70]. Both regularization- and rehearsal-based methods directly train a dense model, which might be over-parameterized even for the union of all tasks [19, 39]. Though several architecture-based methods [51, 57, 64] start with a sparse sub-network from the dense model, they still grow the model size progressively to learn emerging tasks. The aforementioned methods, although striving for greater performance with less forgetting, still introduce significant memory and computation overhead during the whole CL process.

---

*Both authors contributed equally to this work

36th Conference on Neural Information Processing Systems (NeurIPS 2022).

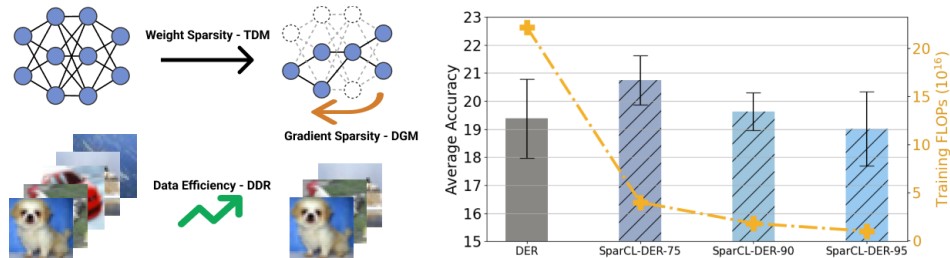

Figure 1: **Left:** Overview of SparCL. SparCL consists of three complementary components: task-aware dynamic masking (TDM) for weight sparsity, dynamic data removal (DDR) for data efficiency, and dynamic gradient masking (DGM) for gradient sparsity. **Right:** SparCL successfully preserves the accuracy and significantly improves efficiency over DER++ [8], one of the SOTA CL methods, with different sparsity ratios on the Split Tiny-ImageNet [16] dataset.

Recently, another stream of work, sparse training [4, 20, 35], has emerged as a new training trend to achieve training acceleration, which embraces the promising training-on-the-edge paradigm. With sparse training, each iteration takes less time with the reduction in computation achieved by sparsity. Inspired by these sparse training methods, under the traditional i.i.d. learning setting, we naturally think about introducing sparse training to the field of CL. A straightforward idea is to directly combine existing sparse training methods, such as SNIP [35], RigL [20], with a rehearsal buffer under the CL setting. However, these methods fail to consider key challenges in CL to mitigate catastrophic forgetting, for example, properly handling transition between tasks. As a result, these sparse training methods, though enhancing training efficiency, cause significant accuracy drop (see Section 5.2). Thus, we would like to explore a general strategy, orthogonal to existing CL methods, that not only leverages the idea of sparse training for efficiency, but also addresses key challenges in CL to preserve (or even improve) accuracy.

In this work, we propose *Sparse Continual Learning* (SparCL), a general framework for cost-effective continual learning, aiming at enabling practical CL on edge devices. As shown in Figure 1 (left), SparCL achieves both learning acceleration and accuracy preservation through the synergy of three aspects: *weight sparsity*, *data efficiency*, and *gradient sparsity*. Specifically, to maintain a small dynamic sparse network during the whole CL process, we develop a novel task-aware dynamic masking (TDM) strategy to keep only important weights for both the current and past tasks, with special consideration during task transitions. Moreover, we propose a dynamic data removal (DDR) scheme, which progressively removes "easy-to-learn" examples from training iterations, which further accelerates the training process and also improves accuracy of CL by balancing current and past data and keeping more informative samples in the buffer. Finally, we provide an additional dynamic gradient masking (DGM) strategy to leverage gradient sparsity for even better efficiency and knowledge preservation of learned tasks, such that only a subset of sparse weights are updated. Figure 1 (right) demonstrates that SparCL successfully preserves the accuracy and significantly improves efficiency over DER++ [8], one of the SOTA CL methods, under different sparsity ratios.

SparCL is simple in concept, compatible with various existing rehearsal-based CL methods, and efficient under practical scenarios. We conduct comprehensive experiments on multiple CL benchmarks to evaluate the effectiveness of our method. We show that SparCL works collaboratively with existing CL methods, greatly accelerates the learning process under different sparsity ratios, and even sometimes improves upon the state-of-the-art accuracy. We also establish competing baselines by combining representative sparse training methods with advanced rehearsal-based CL methods. SparCL again outperforms these baselines in terms of both efficiency and accuracy. Most importantly, we evaluate our SparCL framework on real edge devices to demonstrate the practical potential of our method. We are not aware of any prior CL works that explored this area and considered the constraints of limited resources during training.

In summary, our work makes the following contributions:

- We propose *Sparse Continual Learning* (SparCL), a general framework for cost-effective continual learning, which achieves learning acceleration through the synergy of *weight sparsity*, *data effi-*

*ciency*, and *gradient sparsity*. To the best of our knowledge, our work is the first to introduce the idea of sparse training to enable efficient CL on edge devices. Our code is publicly available*.

- SparCL shows superior performance compared to both conventional CL methods and CL-adapted sparse training methods on all benchmark datasets, leading to at most $23\times$ less training FLOPs and, surprisingly, an $1.7\%$ improvement over SOTA accuracy.

- We evaluate SparCL on a real mobile edge device, demonstrating the practical potential of our method and also encouraging future research on CL on-the-edge. The results indicate that our framework can achieve at most $3.1\times$ training acceleration.

## 2 Related work

### 2.1 Continual Learning

The main focus in continual learning (CL) has been mitigating catastrophic forgetting. Existing methods can be classified into three major categories. *Regularization-based* methods [2, 32, 37, 68] limit updates of important parameters for the prior tasks by adding corresponding regularization terms. While these methods reduce catastrophic forgetting to some extent, their performance deteriorates under challenging settings [40], and on more complex benchmarks [50, 61]. *Rehearsal-based* methods [13, 14, 25] save examples from previous tasks into a small-sized buffer to train the model jointly with the current task. Though simple in concept, the idea of rehearsal is very effective in practice and has been adopted by many state-of-the-art methods [8, 11, 49]. *Architecture-based* methods [42, 51, 57, 59, 63] isolate existing model parameters or assign additional parameters for each task to reduce interference among tasks. As mentioned in Section 1, most of these methods use a dense model without consideration of efficiency and memory footprint, thus are not applicable to resource-limited settings. Our work, orthogonal to these methods, serves as a general framework for making these existing methods efficient and enabling a broader deployment, *e.g.*, CL on edge devices.

A limited number of works explore sparsity in CL, however, for different purposes. Several methods [41, 42, 53, 57] incorporate the idea of weight pruning [24] to allocate a sparse sub-network for each task to reduce inter-task interference. Nevertheless, these methods still reduce the full model sparsity progressively for every task and finally end up with a much denser model. On the contrary, SparCL maintains a sparse network throughout the whole CL process, introducing great efficiency and memory benefits both during training and at the output model. A recent work [15] aims at discovering lottery tickets [21] under CL, but still does not address efficiency. However, the existence of lottery tickets in CL serves as a strong justification for the outstanding performance of our SparCL.

### 2.2 Sparse Training

There are two main approaches for sparse training: fixed-mask sparse training and dynamic sparse training. Fixed-mask sparse training methods [35, 54, 56, 60] first apply pruning, then execute traditional training on the sparse model with the obtained fixed mask. The pre-fixed structure limits the accuracy performance, and the first stage still causes huge computation and memory consumption. To overcome these drawbacks, dynamic mask methods [4, 17, 20, 45, 46] adjust the sparsity topology during training while maintaining low memory footprint. These methods start with a sparse model structure from an untrained dense model, then combine sparse topology exploration at the given sparsity ratio with the sparse model training. Recent work [67] further considers incorporating data efficiency into sparse training for better training accelerations. However, all prior sparse training works are focused on the traditional training setting, while CL is a more complicated and difficult scenario with inherent characteristics not explored by these works. In contrast to prior sparse training methods, our work explores a new learning paradigm that introduces sparse training into CL for efficiency and also addresses key challenges in CL, mitigating catastrophic forgetting.

## 3 Continual Learning Problem Setup

In supervised CL, a model $f_\theta$ learns from a sequence of tasks $\mathcal{D} = \{\mathcal{D}_1, \ldots, \mathcal{D}_T\}$, where each task $\mathcal{D}_t = \{(\boldsymbol{x}_{t,i}, y_{t,i})\}_{i=1}^{n_t}$ consists of input-label pairs, and each task has a disjoint set of classes. Tasks

---

*https://github.com/neu-spiral/SparCL

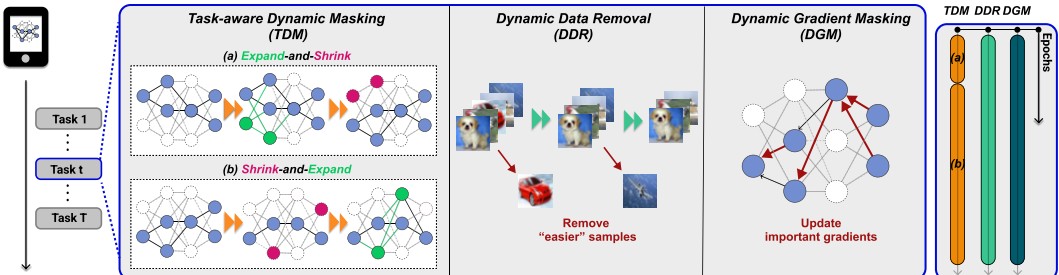

Figure 2: Illustration of the SparCL workflow. Three components work synergistically to improve training efficiency and further mitigate catastrophic forgetting for preserving accuracy.

arrive sequentially, and the model must adapt to them. At the $t$-th step, the model gains access to data from the $t$-th task. However, a small fix-sized rehearsal buffer $\mathcal{M}$ is allowed to save data from prior tasks. At test time, the easiest setting is to assume task identity is known for each coming test example, named task-incremental learning (Task-IL). If this assumption does not hold, we have the more difficult class-incremental learning (Class-IL) setting. In this work, we mainly focus on the more challenging Class-IL setting, and only report Task-IL performance for reference.

The goal of conventional CL is to train a model sequentially that performs well on all tasks at test time. The main evaluation metric is average test accuracy on all tasks. In real-world resource-limited scenarios, we should further consider *training efficiency* of the model. Thus, we measure the performance of the model more comprehensively by including training FLOPs and memory footprint.

## 4 Sparse Continual Learning (SparCL)

Our method, Sparse Continual Learning, is a unified framework composed of three complementary components: *task-aware dynamic masking* for weight sparsity, *dynamic data removal* for data efficiency, and *dynamic gradient masking* for gradient sparsity. The entire framework is shown in Figure 2. We will illustrate each component in detail in this section.

### 4.1 Task-aware Dynamic Masking

To enable cost-effective CL in resource limited scenarios, SparCL is designed to maintain a dynamic structure when learning a sequence of tasks, such that it not only achieves high efficiency, but also intelligently adapts to the data stream for better performance. Specifically, we propose a strategy named *task-aware dynamic masking* (TDM), which dynamically removes less important weights and grows back unused weights for stronger representation power periodically by maintaining a single binary weight mask throughout the CL process. Different from typical sparse training work, which only leverages the weight magnitude [45] or the gradient w.r.t. data from a single training task [20, 67], TDM considers also the importance of weights w.r.t. data saved in the rehearsal buffer, as well as the switch between CL tasks.

Specifically, TDM strategy starts from a randomly initialized binary mask $M_\theta = M_0$, with a given sparsity constraint $\|M_\theta\|_0 / \|\theta\|_0 = 1 - s$, where $s \in [0, 1]$ is the sparsity ratio. Moreover, it makes different intra- and inter-task adjustments to keep a dynamic sparse set of weights based on their continual weight importance (CWI). We summarize the process of task-aware dynamic masking in Algorithm 1 and elaborate its key components in detail below.

**Continual weight importance (CWI).** For a model $f_\theta$ parameterized by $\theta$, the CWI of weight $w \subset \theta$ is defined as follows:

$$\text{CWI}(w) = |w| + \alpha |\frac{\partial \tilde{\mathcal{L}}(\mathcal{D}_t; \theta)}{\partial w}| + \beta |\frac{\partial \mathcal{L}(\mathcal{M}; \theta)}{\partial w}|, \tag{1}$$

where $\mathcal{D}_t$ denotes the training data from the $t$-th task, $\mathcal{M}$ is the current rehearsal buffer, and $\alpha$, $\beta$ are coefficients to control the influence of current and buffered data, respectively. Moreover, $\mathcal{L}$ represents the cross-entropy loss for classification, while $\tilde{\mathcal{L}}$ is the *single-head* [1] version of the cross-entropy loss, which only considers classes from the current task by masking out the logits of other classes.

---

**Algorithm 1:** Task-aware Dynamic Masking (TDM)

---

**Input**: Model weight $\theta$, number of tasks $T$, training epochs of the $t$-th task $K_t$, binary sparse mask $M_\theta$, sparsity ratio $s$, intra-task adjustment ratio $p_{\texttt{intra}}$, inter-task adjustment ratio $p_{\texttt{inter}}$, update interval $\delta k$

**Initialize:** $\theta$, $M_\theta$, s.t. $\|M_\theta\|_0/\|\theta\|_0 = 1 - s$

**for** $t = 1, \ldots, T$ **do**
  **for** $e = 1, \ldots, K_T$ **do**
    **if** $t > 1$ **then**
      /* Inter-task adjustment */
      Expand $M_\theta$ by randomly adding unused weights,
        s.t. $\|M_\theta\|_0/\|\theta\|_0 = 1 - (s - p_{\texttt{inter}})$
      **if** $e = \delta k$ **then**
        Shrink $M_\theta$ by removing the least important weights according to equation 1,
          s.t. $\|M_\theta\|_0/\|\theta\|_0 = 1 - s$
      **end**
    **end**
    **if** $e \bmod \delta k = 0$ **then**
      /* Intra-task adjustment */
      Shrink $M_\theta$ by removing the least important weights according to equation 1,
        s.t. $\|M_\theta\|_0/\|\theta\|_0 = 1 - (s + p_{\texttt{intra}})$
      Expand $M_\theta$ by randomly adding unused weights,
        s.t. $\|M_\theta\|_0/\|\theta\|_0 = 1 - s$
    **end**
    Update $\theta \odot M_\theta$ via backpropagation
  **end**
**end**

---

Intuitively, CWI ensures we keep (1) weights of larger magnitude for output stability, (2) weights important for the current task for learning capacity, and (3) weights important for past data to mitigate catastrophic forgetting. Moreover, inspired by the classification bias in CL [1], we use the single-head cross-entropy loss when calculating the importance score w.r.t. the current task to make the importance estimation more accurate.

**Intra-task adjustment.** When training the $t$-th task, a natural assumption is that the data distribution is consistent inside the task. As a result, we would like to update the sparse model in a relatively stable way while keeping its flexibility. Thus, in Algorithm 1, we choose to update the sparsity mask $M_\theta$ in a *shrink-and-expand* way every $\delta k$ epochs. We first remove $p_{\texttt{intra}}$ of the weights of least CWI to retain learned knowledge so far. Then we randomly select unused weights to recover the learning capacity for the model and keep the sparsity ratio $s$ unchanged.

**Inter-task adjustment.** When tasks switch, on the contrary, we assume that the data distribution shifts immediately. Ideally, we would like the model to keep the knowledge learned from old tasks as much as possible, and to have enough learning capacity to accommodate the new task. Thus, instead of the shrink-and-expand strategy for intra-task adjustment, we follow an *expand-and-shrink* scheme. Specifically, at the beginning of the $(t + 1)$-th task, we expand the sparse model by randomly adding a proportion of $p_{\texttt{inter}}$ unused weights. Intuitively, the additional learning capacity facilitates fast adoption of new knowledge and reduces interference with learned knowledge. We allow our model to have smaller sparsity (*i.e.*, larger learning capacity) temporarily for the first $\delta k$ epochs as a warm-up period, and then remove the $p_{\texttt{inter}}$ weights with least CWI, following the same process in the intra-task case, to satisfy the sparsity constraint.

## 4.2 Dynamic Data Removal

In addition to weight sparsity, decreasing the amount of training data can be directly translated into training time savings. Thus, we would also like to explore data efficiency to reduce the training workload. Some prior CL works select informative examples to construct the rehearsal buffer [3, 6, 65]. However, their main purpose is not training acceleration; thus, they either introduce excessive computational cost or consider different problem settings. By considering the features of CL, we

present a simple yet effective strategy, *dynamic data removal* (DDR), to reduce training data for further acceleration.

We measure the importance of each training example by the occurrence of misclassification [55, 67] during CL. In TDM, the sparse structure of our model updates periodically every $\delta k$ epochs, so we align our data removal process with weight mask updates for further efficiency and training stability. In Section 4.1, we have partitioned the training process for the $t$-th task into $N_t = K_t/\delta k$ stages based on the dynamic mask update. Therefore, we gradually remove training data at the end of $i$-th stage, based on the following policy: 1) Calculate the total number of misclassifications $f_i(x_j)$ for each training example during the $i$-th stage. 2) Remove a proportion of $\rho_i$ training samples with the least number of misclassifications. Although our main purpose is to keep the "harder" examples to learn to consolidate the sparse model, we can get further benefits for better CL results. First, the removal of "easier" examples increases the probability that "harder" examples to be saved to the rehearsal buffer, given the common strategy, *e.g.* reservoir sampling [14], to buffer examples. Thus, we construct a more informative buffer in a implicit way without heavy computation. Moreover, since the buffer size is much smaller than the training set size of each task, the data from the buffer and the new task is highly imbalanced; dynamic data removal also relieves this data imbalance issue.

Formally, we set the data removal proportion for each task as $\rho \in [0, 1]$, and a cutoff stage, such that:

$$\sum_{i=1}^{\texttt{cutoff}} \rho_i = \rho, \qquad \sum_{i=\texttt{cutoff}+1}^{N_k} \rho_i = 0 \tag{2}$$

The cutoff stage controls the trade-off between efficiency and accuracy: when we set the cutoff stage earlier, we reduce the training time for all the following stages; however, when the cutoff stage is set too early, the model might underfit the removed training data. Note that when we set $\rho_i = 0$ for all $i = 1, 2, \ldots, N_t$ and $\texttt{cutoff} = N_t$, we simply recover the vanilla setting without any data efficiency considerations. In our experiments, we assume $\rho_i = \rho/\texttt{cutoff}$, i.e., removing equal proportion of data at the end of every stage, for simplicity. We also conduct a comprehensive exploration study of $\rho$ and the selection of the cutoff stage in Section 5.3 and Appendix B.3.

## 4.3 Dynamic Gradient Masking

With TDM and DDR, we can already achieve weight efficiency and data efficiency during training. To further boost training efficiency, we explore gradient sparsity and propose dynamic gradient masking (DGM) for CL. Our method focuses on reducing computational costs by only applying the most important gradients onto the corresponding unpruned model parameters via a gradient mask. The gradient mask is also dynamically updated along with the weight mask defined in Section 4.1. Intuitively, while targeting for better training efficiency, DGM also promotes the preservation of past knowledge by preventing a fraction of weights from updating.

Formally, our goal here is to find a subset of unpruned parameters (or, equivalently, a gradient mask $M_G$) to update over multiple training iterations. For a model $f_\theta$ parameterized by $\theta$, we have the corresponding gradient matrix $G$ calculated during each iteration. To prevent the pruned weights from updating, the weight mask $M_\theta$ will be applied onto the gradient matrix $G$ as $G \odot M_\theta$ during backpropagation. Besides the gradients of pruned weights, we in addition consider to remove less important gradient coefficients for faster training. To achieve this, we introduce the continual gradient importance (CGI) based on the CWI to measure the importance of weight gradients:

$$\text{CGI}(w) = \alpha |\frac{\partial \tilde{\mathcal{L}}(\mathcal{D}_t; \theta)}{\partial w}| + \beta |\frac{\partial \mathcal{L}(\mathcal{M}; \theta)}{\partial w}|. \tag{3}$$

We remove a proportion $q$ of non-zero gradients from $G$ with less importance measured by CGI and we have $\|M_G\|_0/\|\theta\|_0 = 1 - (s + q)$. The gradient mask $M_G$ is then applied onto the gradient matrix $G$. During the entire training process, the gradient mask $M_G$ is updated with a fixed interval.

# 5 Experiment

## 5.1 Experiment Setting

**Datasets.** We evaluate our SparCL on two representative CL benchmarks, Split CIFAR-10 [33] and Split Tiny-ImageNet [16] to verify the efficacy of SparCL. In particular, we follow [8, 68] by

Table 1: Comparison with CL methods. SparCL consistently improves training efficiency of the corresponding CL methods while preserves (or even improves) accuracy on both class- and task-incremental settings.

| Method | Sparsity | Buffer size | Split CIFAR-10 | | | Split Tiny-ImageNet | | |
|---|---|---|---|---|---|---|---|---|
| | | | Class-IL ($\uparrow$) | Task-IL ($\uparrow$) | FLOPs Train $\times 10^{15}$ ($\downarrow$) | Class-IL ($\uparrow$) | Task-IL ($\uparrow$) | FLOPs Train $\times 10^{16}$ ($\downarrow$) |
| EWC [32] | 0.00 | – | $19.49_{\pm 0.12}$ | $68.29_{\pm 3.92}$ | 8.3 | $7.58_{\pm 0.10}$ | $19.20_{\pm 0.31}$ | 13.3 |
| LwF [37] | | | $19.61_{\pm 0.05}$ | $63.29_{\pm 2.35}$ | 8.3 | $8.46_{\pm 0.22}$ | $15.85_{\pm 0.58}$ | 13.3 |
| PackNet [42] | $0.50^{\dagger}$ | – | - | $93.73_{\pm 0.55}$ | 5.0 | – | $61.88_{\pm 1.01}$ | 7.3 |
| LPS [57] | | | - | $94.50_{\pm 0.47}$ | 5.0 | – | $63.37_{\pm 0.83}$ | 7.3 |
| A-GEM [13] | 0.00 | 200 | $20.04_{\pm 0.34}$ | $83.88_{\pm 1.49}$ | 11.1 | $8.07_{\pm 0.08}$ | $22.77_{\pm 0.03}$ | 17.8 |
| iCaRL [50] | | | $49.02_{\pm 3.20}$ | $88.99_{\pm 2.13}$ | 11.1 | $7.53_{\pm 0.79}$ | $28.19_{\pm 1.47}$ | 17.8 |
| FDR [5] | | | $30.91_{\pm 2.74}$ | $91.01_{\pm 0.68}$ | 13.9 | $8.70_{\pm 0.19}$ | $40.36_{\pm 0.68}$ | 22.2 |
| ER [14] | | | $44.79_{\pm 1.86}$ | $91.19_{\pm 0.94}$ | 11.1 | $8.49_{\pm 0.16}$ | $38.17_{\pm 2.00}$ | 17.8 |
| DER++ [8] | | | $64.88_{\pm 1.17}$ | $91.92_{\pm 0.60}$ | 13.9 | $10.96_{\pm 1.17}$ | $40.87_{\pm 1.16}$ | 22.2 |
| SparCL-ER$_{75}$ | 0.75 | 200 | $46.89_{\pm 0.68}$ | $92.02_{\pm 0.72}$ | 2.0 | $8.98_{\pm 0.38}$ | $39.14_{\pm 0.85}$ | 3.2 |
| SparCL-DER++$_{75}$ | | | $66.30_{\pm 0.98}$ | $94.06_{\pm 0.45}$ | 2.5 | $12.73_{\pm 0.40}$ | $42.06_{\pm 0.73}$ | 4.0 |
| SparCL-ER$_{90}$ | 0.90 | | $45.81_{\pm 1.05}$ | $91.49_{\pm 0.47}$ | 0.9 | $8.67_{\pm 0.41}$ | $38.79_{\pm 0.39}$ | 1.4 |
| SparCL-DER++$_{90}$ | | | $65.79_{\pm 1.33}$ | $93.73_{\pm 0.24}$ | 1.1 | $12.27_{\pm 1.06}$ | $41.17_{\pm 1.31}$ | 1.8 |
| SparCL-ER$_{95}$ | 0.95 | | $44.59_{\pm 0.23}$ | $91.07_{\pm 0.64}$ | 0.5 | $8.43_{\pm 0.09}$ | $38.20_{\pm 0.46}$ | 0.8 |
| SparCL-DER++$_{95}$ | | | $65.18_{\pm 1.25}$ | $92.97_{\pm 0.37}$ | 0.6 | $10.76_{\pm 0.62}$ | $40.54_{\pm 0.98}$ | 1.0 |
| A-GEM [13] | 0.00 | 500 | $22.67_{\pm 0.57}$ | $89.48_{\pm 1.45}$ | 11.1 | $8.06_{\pm 0.04}$ | $25.33_{\pm 0.49}$ | 17.8 |
| iCaRL [50] | | | $47.55_{\pm 3.95}$ | $88.22_{\pm 2.62}$ | 11.1 | $9.38_{\pm 1.53}$ | $31.55_{\pm 3.27}$ | 17.8 |
| FDR [5] | | | $28.71_{\pm 3.23}$ | $93.29_{\pm 0.59}$ | 13.9 | $10.54_{\pm 0.21}$ | $49.88_{\pm 0.71}$ | 22.2 |
| ER [14] | | | $57.74_{\pm 0.27}$ | $93.61_{\pm 0.27}$ | 11.1 | $9.99_{\pm 0.29}$ | $48.64_{\pm 0.46}$ | 17.8 |
| DER++ [8] | | | $72.70_{\pm 1.36}$ | $93.88_{\pm 0.50}$ | 13.9 | $19.38_{\pm 1.41}$ | $51.91_{\pm 0.68}$ | 22.2 |
| SparCL-ER$_{75}$ | 0.75 | 500 | $60.80_{\pm 0.22}$ | $93.82_{\pm 0.32}$ | 2.0 | $10.48_{\pm 0.29}$ | $50.83_{\pm 0.69}$ | 3.2 |
| SparCL-DER++$_{75}$ | | | $74.09_{\pm 0.84}$ | $95.19_{\pm 0.34}$ | 2.5 | $20.75_{\pm 0.88}$ | $52.19_{\pm 0.43}$ | 4.0 |
| SparCL-ER$_{90}$ | 0.90 | | $59.34_{\pm 0.97}$ | $93.33_{\pm 0.10}$ | 0.9 | $10.12_{\pm 0.53}$ | $49.46_{\pm 1.22}$ | 1.4 |
| SparCL-DER++$_{90}$ | | | $73.42_{\pm 0.95}$ | $94.82_{\pm 0.23}$ | 1.1 | $19.62_{\pm 0.67}$ | $51.93_{\pm 0.36}$ | 1.8 |
| SparCL-ER$_{95}$ | 0.95 | | $57.75_{\pm 0.45}$ | $92.73_{\pm 0.34}$ | 0.5 | $9.91_{\pm 0.17}$ | $48.57_{\pm 0.50}$ | 0.8 |
| SparCL-DER++$_{95}$ | | | $72.14_{\pm 0.78}$ | $94.39_{\pm 0.15}$ | 0.6 | $19.01_{\pm 1.32}$ | $51.26_{\pm 0.78}$ | 1.0 |

$^{\dagger}$PackNet and LPS actually have a decreased sparsity after learning every task, we use 0.50 to roughly represent the average sparsity.

splitting CIFAR-10 and Tiny-ImageNet into 5 and 10 tasks, each of which consists of 2 and 20 classes respectively. Dataset licensing information can be found in Appendix A.

**Comparing methods.** We select several CL methods including regularization-based (EWC [32], LwF [37]), architecture-based (PackNet [42], LPS [57]), and rehearsal-based (A-GEM [13], iCaRL [44], FDR [5], ER [14], DER++ [8]) methods. Note that PackNet and LPS are only compatible with task-incremental learning. We also adapt representative sparse training methods (SNIP [35], RigL [20]) to the CL setting by combining them with DER++ (SNIP-DER++, RigL-DER++).

**Variants of our method.** To show the generality of SparCL, we combine it with DER++ (one of the SOTA CL methods), and ER (simple and widely-used) as *SparCL-DER++* and *SparCL-ER*, respectively. We also vary the weight sparsity ratio $(0.75, 0.90, 0.95)$ of SparCL for a comprehensive evaluation.

**Evaluation metrics.** We use the average accuracy on all tasks to evaluate the performance of the final model. Moreover, we measure the training FLOPs [20], and memory footprint [67] (including feature map pixels and model parameters during training) to demonstrate the efficiency of each method. Please see Appendix B.1 for detailed definitions of these metrics.

**Experiment details.** For fair comparison, we strictly follow the settings in prior CL work [8, 29]. We set the per task training epochs to 50 and 100 for Split CIFAR-10 and Tiny-ImageNet, respectively, with a batch size of 32. For the model architecture, we follow [8, 50] and adopt the ResNet-18 [26] without any pre-training. We also use the best hyperparameter setting reported in [8, 57] for CL methods, and in [20, 35] for CL-adapted sparse training methods. For SparCL and its competing CL-adapted sparse training methods, we adopt a uniform sparsity ratio for all convolutional layers. Please see Appendix B for further details.

### 5.2 Main Results

**Comparison with CL methods.** Table 1 summarizes the results on Split CIFAR-10 and Tiny-ImageNet, under both class-incremental (Class-IL) and task-incremental (Task-IL) settings. From Table 1, we can clearly tell that SparCL significantly improves upon ER and DER++, while also

Table 2: Comparison with CL-adapted sparse training methods. All methods are combined with DER++ with a 500 buffer size. SparCL outperforms all methods in both accuracy and training efficiency, under all sparsity ratios. All three methods here can save $20\% \sim 51\%$ memory footprint, please see Appendix B.2 for details.

| Method | Spasity | Split CIFAR-10 | | Split Tiny-ImageNet | |
| --- | --- | --- | --- | --- | --- |
| | | Class-IL ($\uparrow$) | FLOPs Train $\times 10^{15}$ ($\downarrow$) | Class-IL ($\uparrow$) | FLOPs Train $\times 10^{16}$ ($\downarrow$) |
| DER++ [8] | 0.00 | $72.70_{\pm 1.36}$ | 13.9 | $19.38_{\pm 1.41}$ | 22.2 |
| SNIP-DER++ [35] | | $69.82_{\pm 0.72}$ | 1.6 | $16.13_{\pm 0.61}$ | 2.5 |
| RigL-DER++ [20] | 0.90 | $69.86_{\pm 0.59}$ | 1.6 | $18.36_{\pm 0.49}$ | 2.5 |
| SparCL-DER++$_{90}$ | | $73.42_{\pm 0.95}$ | 1.1 | $19.62_{\pm 0.67}$ | 1.8 |
| SNIP-DER++ [35] | | $66.07_{\pm 0.91}$ | 0.9 | $14.76_{\pm 0.52}$ | 1.5 |
| RigL-DER++ [20] | 0.95 | $66.53_{\pm 1.13}$ | 0.9 | $15.88_{\pm 0.63}$ | 1.5 |
| SparCL-DER++$_{95}$ | | $72.14_{\pm 0.78}$ | 0.6 | $19.01_{\pm 1.32}$ | 1.0 |

Table 3: Ablation study on Split-CIFAR10 with 0.75 sparsity ratio. All components contributes to the overall performance, in terms of both accuracy and efficiency (training FLOPs and memory footprint).

| TDM | DDR | DGM | Class-IL ($\uparrow$) | FLOPs Train $\times 10^{15}$ ($\downarrow$) | Memory Footprint ($\downarrow$) |
| --- | --- | --- | --- | --- | --- |
| ✗ | ✗ | ✗ | 72.70 | 13.9 | 247MB |
| ✓ | ✗ | ✗ | 73.37 | 3.6 | 180MB |
| ✓ | ✓ | ✗ | 73.80 | 2.8 | 180MB |
| ✓ | ✗ | ✓ | 73.97 | 3.3 | 177MB |
| ✓ | ✓ | ✓ | **74.09** | **2.5** | **177MB** |

outperforming other CL baselines, in terms of training efficiency (measured in FLOPs). With higher sparsity ratio, SparCL leads to fewer training FLOPs. Notably, SparCL achieves $23\times$ training efficiency improvement upon DER++ with a sparsity ratio of 0.95. On the other hand, our framework also improves the average accuracy of ER and DER++ consistently under all cases with a sparsity ratio of 0.75 and 0.90, and only a slight performance drop when sparsity gets larger as 0.95. In particular, SparCL-DER++ with 0.75 sparsity ratio sets new SOTA accuracy, with all buffer sizes under both benchmarks. The outstanding performance of SparCL indicates that our proposed strategies successfully preserve accuracy by further mitigating catastrophic forgetting with a much sparser model. Moreover, the improvement that SparCL brings to two different existing CL methods shows the generalizability of SparCL as a unified framework, *i.e.*, it has the potential to be combined with a wide array of existing methods.

We also take a closer look at PackNet and LPS, which also leverage the idea of sparsity to split the model by different tasks, a different motivation from training efficiency. Firstly, they are only compatible with the Task-IL setting, since they leverage task identity at both training and test time. Moreover, the model sparsity of these methods reduces with the increasing number of tasks, which still leads to much larger overall training FLOPs than that of SparCL. This further demonstrates the importance of keeping a sparse model without permanent expansion throughout the CL process.

**Comparison with CL-adapted sparse training methods.** Table 2 shows results under the more difficult Class-IL setting. SparCL outperforms all CL-adapted sparse training methods in both accuracy and training FLOPs. The performance gap between SparCL-DER++ and other methods gets larger with a higher sparsity. SNIP- and RigL-DER++ achieve training acceleration at the cost of compromised accuracy, which suggests that keeping accuracy is a non-trivial challenge for existing sparse training methods under the CL setting. SNIP generates the static initial mask after network initialization which does not consider the structure suitability among tasks. Though RigL adopts a dynamic mask, the lack of task-awareness prevents it from generalizing well to the CL setting.

### 5.3 Effectiveness of Key Components

**Ablation study.** We provide a comprehensive ablation study in Table 3 using SparCL-DER++ with 0.75 sparsity on Split CIFAR10. Table 3 demonstrates that all components of our method contribute to both efficiency and accuracy improvements. Comparing rows 1 and 2, we can see that the majority of FLOPs decrease results from TDM. Interestingly, TDM leads to an increase in accuracy, indicating

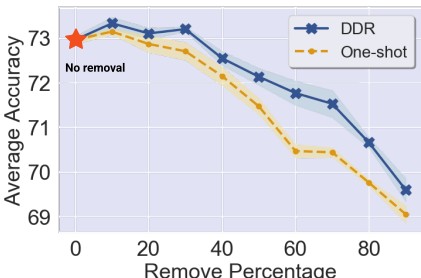

Figure 3: Comparison between DDR and One-shot [67] data removal strategy w.r.t. different data removal proportion $\rho$. DDR outperforms One-shot and also achieves better accuracy when $\rho \leq 30\%$.

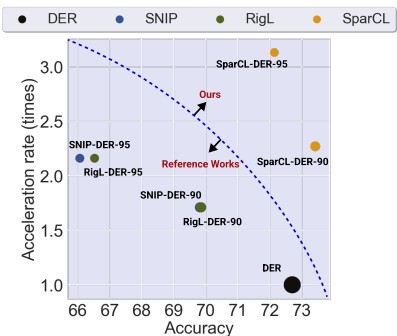

Figure 4: Comparison with CL-adapted sparse training methods in training acceleration rate and accuracy results. The radius of circles are measured by memory footprint.

TDM generates a sparse model that is even more suitable for learning all tasks than then full dense model. Comparing rows 2 and 3, we can see that DDR indeed further accelerates training by removing less informative examples. As discussed in Section 4.2, when we remove a certain number of samples (30% here), we achieve a point where we keep as much informative samples as we need, and also balance the current and buffered data. Comparing rows 2 and 4, DGM reduce both training FLOPs and memory footprint while improve the performance of the network. Finally, the last row demonstrates the collaborative performance of all components. We also show the same ablation study with 0.90 sparsity in Appendix B.4 for reference. Details can be found in Appendix B.1.

**Exploration on DDR.** To understand the influence of the data removal proportion $\rho$, and the `cutoff` stage for each task, we show corresponding experiment results in Figure 3 and Appendix B.3, respectively. In Figure 3, we fix `cutoff` $= 4$, *i.e.*, gradually removing equal number of examples every 5 epochs until epoch 20, and vary $\rho$ from $10\%$ to $90\%$. We also compare DDR with One-shot removal strategy [67], which removes all examples at once at `cutoff`. DDR outperforms One-shot consistently with different $\rho$ in average accuracy. Also note that since DDR removes the examples gradually before the `cutoff` stage, DDR is more efficient than One-shot. When $\rho \leq 30\%$, we also observe increased accuracy of DDR compared with the baseline without removing any data. When $\rho \geq 40\%$, the accuracy gets increasingly lower for both strategies. The intuition is that when DDR removes a proper amount of data, it removes redundant information while keeps the most informative examples. Moreover, as discussed in Section 4.2, it balances the current and buffered data, while also leave informative samples in the buffer. When DDR removes too much data, it will also lose informative examples, thus the model has not learned these examples well before removal.

**Exploration on DGM.** We test the efficacy of DGM at different sparsity levels. Detailed exploratory experiments are shown in Appendix B.5 for reference. The results indicate that by setting the proportion $q$ within an appropriate range, DGM can consistently improve the accuracy performance regardless of the change of weight sparsity.

### 5.4 Mobile Device Results

The training acceleration results are measured on the CPU of an off-the-shelf Samsung Galaxy S20 smartphone, which has the Qualcomm Snapdragon 865 mobile platform with a Qualcomm Kryo 585 Octa-core CPU. We run each test on a batch of 32 images to denote the training speed. The detail of on-mobile compiler-level optimizations for training acceleration can be found in Appendix C.1.

The acceleration results are shown in Figure 4. SparCL can achieve approximately $3.1\times$ and $2.3\times$ training acceleration with 0.95 sparsity and 0.90 sparsity, respectively. Besides, our framework can also save 51% and 48% memory footprint when the sparsity is 0.95 and 0.90. Furthermore, the obtained sparse models save the storage consumption by using compressed sparse row (CSR) storage and can be accelerated to speed up the inference on-the-edge. We provide on-mobile inference acceleration results in Appendix C.2.

# 6 Conclusion

This paper presents a unified framework named SparCL for efficient CL that achieves both learning acceleration and accuracy preservation. It comprises three complementary strategies: task-aware dynamic masking for weight sparsity, dynamic data removal for data efficiency, and dynamic gradient masking for gradient sparsity. Extensive experiments on standard CL benchmarks and real-world edge device evaluations demonstrate that our method significantly improves upon existing CL methods and outperforms CL-adapted sparse training methods. We discuss the limitations and potential negative social impacts of our method in Sections 7 and 8, respectively.

# 7 Limitations

One limitation of our method is that we assume a rehearsal buffer is available throughout the CL process. Although the assumption is widely-accepted, there are still situations that a rehearsal buffer is not allowed. However, as a framework targeting for efficiency, our work has the potential to accelerate all types of CL methods. For example, simply removing the terms related to rehearsal buffer in equation 1 and equation 3 could serve as a naive variation of our method that is compatible with other non-rehearsal methods. It is interesting to further improve SparCL to be more generic for all kinds of CL methods. Moreover, the benchmarks we use are limited to vision domain. Although using vision-based benchmarks has been a common practice in the CL community, we believe evaluating our method, as well as other CL methods, on datasets from other domains such as NLP will lead to a more comprehensive and reliable conclusion. We will keep track of newer CL benchmarks from different domains and further improve our work correspondingly.

# 8 Potential Negative Societal Impact

Although SparCL is a general framework to enhance efficiency for various CL methods, we still need to be aware of its potential negative societal impact. For example, we need to be very careful about the trade-off between accuracy and efficiency when using SparCL. If one would like to pursue efficiency by setting the sparsity ratio too high, then even SparCL will result in significant accuracy drop, since the over-sparsified model does not have enough representation power. Thus, we should pay much attention when applying SparCL on accuracy-sensitive applications such as healthcare [66]. Another example is that, SparCL as a powerful tool to make CL methods efficient, can also strengthen models for malicious applications [7]. Therefore, we encourage the community to come up with more strategies and regulations to prevent malicious use of artificial intelligence.

# 9 Acknowledgement

The authors gratefully acknowledge support by the National Science Foundation under grants CCF-1937500, CCF-1919117 and CNS-2112471.

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
