# A    Dataset Licensing Information

- CIFAR-10 [33] is licensed under the MIT license.
- The licensing information of Tiny-ImageNet [34] is not available. However, the data is available for free to researchers for non-commercial use.

# B    Additional Experiment Details and Results

We set $\alpha = 0.5, \beta = 1$ in equation 1 and equation 3. We also set $\delta k = 5$, $p_{\text{inter}} = 0.01$, $p_{\text{intra}} = 0.005$. We also match different weight sparsity with gradient sparsity for best performance. We sample 20% data from Split CIFAR-10 training set for validation, and we use grid-search on this validation set to help us select the mentioned best hyperparameters. We use the same set of hyperparameters for both datasets. For accurate evaluation, we repeat each experiments 3 times using different random seeds and report the average performance. During our experiments, we adopt unstructured sparsity type and uniform sparsity ratio $(0.75, 0.90, 0.95)$ for all convolutional layers in the models.

## B.1    Evaluation Metrics Explanation

**Training FLOPs** The FLOPs of a single forward pass is calculated by taking the sum of the number of multiplications and additions in each layer $l$ for a given layer sparsity $s_l$. Each iteration in the training process is composed of two phases, i.e., the forward propagation and backward propagation.

The goal of the forward pass is to calculate the loss of the current set of parameters on a given batch of data. It can be formulated as $a_l = \sigma(z_l) = \sigma(w_l * a_{l-1} + b_l)$ for each layer $l$ in the model. Here, $w$, $b$, and $z$ represent the weights, biases, and output before activation, respectively; $\sigma(.)$ denotes the activation function; $a$ is the activations; $*$ means convolution operation. The formulation indicates that the layer activations are calculated in sequence using the previous activations and the parameters of the layer. Activation of layers are stored in memory for the backward pass.

As for the backward propogation, the objective is to back-propagate the error signal while calculating the gradients of the parameters. The two main calculation steps can be represented as:

$$\delta_l = \delta_{l+1} * \text{rotate}180°(w_l) \odot \sigma'(z_l),$$  (4)
$$G_l = a_{l-1} * \delta_l,$$  (5)

where $\delta_l$ is the error associated with the layer $l$, $G_l$ denotes the gradients, $\odot$ represents Hadamard product, $\sigma'(.)$ denotes the derivative of activation, and rotate$180°(.)$ means rotating the matrix by $180°$ is the matrix transpose operation. During the backward pass, each layer $l$ calculates two quantities, i.e., the gradient of the activations of the previous layer and the gradient of its parameters. Thus, the backward passes are counted as **twice** the computation expenses of the forward pass [20]. We omit the FLOPs needed for batch normalization and cross entropy. In our work, the total FLOPs introduced by TDM, DDR, and DGM on split CIFAR-10 is approximately $4.5 \times 10^9$ which is less than $0.0001\%$ of total training FLOPs. For split Tiny-ImageNet, the total FLOPs of them is approximately $1.8 \times 10^{10}$, which is also less than $0.0001\%$ of total training FLOPs. Therefore, the computation introduced by TDM, DDR, and DGM is negligible.

**Memory Footprints** Following works [10, 67], the definition of memory footprints contain two parts: 1) activations (feature map pixels) during training phase, and 2) model parameters during training phase. For experiments, activations, model weights, and gradients are stored in 32-bit floating-point format for training. The memory footprint results are calculated with an approximate summation of them.

## B.2    Details of Memory Footprint

The memory footprint is composed of three parts: activations, model weights, and gradients. They are all represented as $b_w$-bit numbers for training.

The number of activations in the model is the sum of the activations in each layer. Suppose that the output feature of the $l$-th layer with a batch size of $B$ is represented as $a_l \in \mathcal{R}^{B \times O_l \times H_l \times W_l}$, where

$O_l$ is the number of channels and $H_l \times W_l$ is the feature size. The total number of activations of the model is thus $B \sum_l O_l H_l W_l$.

As for the model weights, our SparCL training a sparse model with a sparsity ratio $s \in [0, 1]$ from scratch. The sparse model is obtained from a dense model with a total of $N$ weights. A higher value of $s$ indicates fewer non-zero weights in the sparse model. Compressed sparse row (CSR) format is commonly used for sparse storage, which greatly reduces the number of indices need to be stored for sparse matrices. As our SparCL adopt only one sparsity type and we use a low-bit format to store the indices, we omit the indices storage here. Therefore, the memory footprint for model representation is $(1 - s)Nb_w$.

Similar calculations can be applied for the gradient matrix. Besides the sparsity ratio $s$, additional $q$ gradients are masked out from the gradient matrix, resulting a sparsity ratio $s + q$. Therefore, the storage of gradients can be approximated as $(1 - (s + q))Nb_w$.

Combining the activations, model representation, and gradients, the total memory footprint in SparCL can be represented as $(2B \sum_l O_l H_l W_l + (1 - s)N + (1 - (s + q))N)b_w$.

DDR requires store indices for the easier examples during the training process. The number of training examples for Split CIFAR-10 and Split Tiny-ImageNet on each task is 10000. In our work, we only need about 3KB (remove 30% training data) for indices storage (in the int8 format) and the memory cost is negligible compared with the total memory footprint.

## B.3 Effect of Cutoff Stage

Table A1: Effect of `cutoff`.

| `cutoff` | 1 | 2 | 3 | 4 | 5 | 6 | 7 | 8 | 9 |
|---|---|---|---|---|---|---|---|---|---|
| Class-IL ($\uparrow$) | 71.54 | 72.38 | 72.74 | 73.20 | 73.10 | 73.32 | 73.27 | 73.08 | 73.23 |

To evaluate the effect of the `cutoff` stage, we use the same setting as in Figure 3 by setting the sparsity ratio to 0.90. We keep the data removal proportion $\rho = 30\%$, and only change `cutoff`. Table A1 shows the relationship between `cutoff` and the Class-IL average accuracy. Note that from the perspective of efficiency, we would like the `cutoff` stage as early as possible, so that the remaining epochs will have less examples. However, from Table A1, we can see that if we set it too early, *i.e.*, `cutoff` $\leq 3$, the accuracy drop is significant. This indicate that even for the "easy-to-learn" examples, removing them too early results in underfitting. As a balance point between accuracy and efficiency, we choose `cutoff` $= 4$ in our final version.

## B.4 Supplementary Ablation Study

Table A2: Ablation study on Split-CIFAR10 with 0.90 sparsity.

| TDM | DDR | DGM | Class-IL ($\uparrow$) | FLOPs Train $\times 10^{15}$ ($\downarrow$) | Memory Footprint ($\downarrow$) |
|---|---|---|---|---|---|
| ✗ | ✗ | ✗ | 72.70 | 13.9 | 247MB |
| ✓ | ✗ | ✗ | 72.98 | 1.6 | 166MB |
| ✓ | ✓ | ✗ | 73.20 | 1.2 | 166MB |
| ✓ | ✗ | ✓ | 73.30 | 1.5 | 165MB |
| ✓ | ✓ | ✓ | **73.42** | **1.1** | **165MB** |

Similar to Table 3, we show ablation study with 0.90 sparsity ratio in Table A2. Under a larger sparsity ratio, the conclusion that all components contribute to the final performance still holds. However, we can observe that the accuracy increase that comes from DDR and DGM is less than what we show in Table 3. We assume that larger sparsity ratio makes it more difficult for the model to retain good accuracy in CL. Similar results has also been observed in [67] under the usual i.i.d. learning setting.

Table A3: Ablation study of the gradient sparsity ratio on Split-CIFAR10.

| weight sparsity | gradient sparsity | Class-IL (↑) | FLOPs Train ×10$^{15}$ (↓) | Memory Footprint (↓) |
|---|---|---|---|---|
| 0.75 | 0.78 | 74.08 | 3.4 | 178MB |
| 0.75 | 0.80 | 73.97 | 3.3 | 177MB |
| 0.75 | 0.82 | 73.79 | 3.3 | 177MB |
| 0.75 | 0.84 | 73.26 | 3.2 | 176MB |
| 0.90 | 0.91 | 73.33 | 1.6 | 166MB |
| 0.90 | 0.92 | 73.30 | 1.5 | 165MB |
| 0.90 | 0.93 | 72.64 | 1.5 | 165MB |

## B.5   Exploration on DGM

We conduct further experiments to demonstrate the influence of gradient sparsity, and the results are shown in Table A4. There are two sets of the experiments with different weight sparsity settings: 0.75 and 0.90. Within each set of the experiments (the weight sparsity is fixed), we vary the gradient sparsity values. From the results we can see that increasing the gradient sparsity can decrease the FLOPs and memory footprint. However, the accuracy performance degrades more obvious when the gradient sparsity is too much for the weight sparsity. The results indicate that suitable gradient sparsity setting can bring further efficiency to the training process while boosting the accuracy performance. In the main results, the gradient sparsity is set as 0.80 for 0.75 weight sparsity, and set as 0.92 for 0.90 weight sparsity.

# C   On-Mobile Compiler Optimizations and Inference Results

## C.1   Compiler Optimizations

Each iteration in the training process is composed of two phases, i.e., the forward propagation and backward propagation. Prior works [18, 23, 27, 28, 30, 36, 38, 69] have proved that sparse weight matrices (tensors) can provide inference acceleration via reducing the number of multiplications in convolution operation. Therefore, the forward propagation phase, which is the same as inference, can be accelerated by the sparsity inherently. As for backward pass, both of the calculation steps are based on convolution, i.e., matrix multiplication. Equation 4 uses sparse weight matrix (tensor) as the operand, thus can be accelerated in the same way as the forward propagation. Equation 5 allows a sparse output result since the gradient matrix is also sparse. Thus, both two steps have reduced computations, which are roughly proportional to the sparsity ratio, providing the acceleration for the backward propagation phase.

Compiler optimizations are used to accelerate the inference in prior works [22, 47, 62]. In this work, we extend the compiler optimization techniques for accelerating the forward and backward pass during training on the edge devices. Our compiler optimizations are general, support both sparse model training and inference accelerations on mobile platforms. The optimizations include 1) the supports for sparse models; 2) an auto-tuning process to determine the best-suited configurations of parameters for different mobile CPUs. The details of our compiler optimizations are presented as follows.

### C.1.1   Supports for Sparse Models

Our framework supports sparse model training and inference accelerations with unstructured pruning. For the sparse (pruned) model, the framework first compacts the model storage with a compression format called Compressed Sparse Row (CSR) format, and then performs computation reordering to reduce the branches within each thread and eliminates the load imbalance among threads.

A row reordering optimization is also included to further improve the regularity of the weight matrix. After this reordering, the continuous rows with identical or similar numbers of non-zero weights are processed by multi-threads simultaneously, thus eliminating thread divergence and achieving load balance. Each thread processes more than one rows, thus eliminating branches and improving instruction-level parallelism. Moreover, a similar optimization flow (i.e., model compaction and

computation reorder and other optimizations) is employed to support all compiler optimizations for sparsity as PatDNN [47].

### C.1.2 Auto-Tuning for Different Mobile CPUs

During DNN sparse training and inference execution, there are many tuning parameters, e.g., matrix tiling sizes, loop unrolling factors, and data placement on memory, that influence the performance. It is hard to determine the best-suited configuration of these parameters manually. To alleviate this problem, our compiler incorporates an auto-tuning approach for sparse (pruned) models. The Genetic Algorithm is leveraged to explore the best-suited configurations automatically. It starts the parameter search process with an arbitrary number of chromosomes and explores the parallelism better. Acceleration codes for different DNN models and different mobile CPUs can be generated efficiently and quickly through this auto-tuning process.

### C.2 Inference Acceleration Results On Mobile

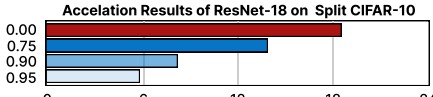
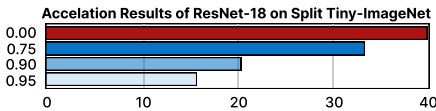

Figure 5: Inference results of sparse models obtained from SparCL under different sparsity ratio compared with dense models obtained from traditional CL methods (sparsity ratio 0.00).

Besides accelerating the training process, SparCL also possesses the advantages of providing a sparse model as the output for faster inference. To demonstrate this, we show the inference acceleration results of SparCL with different sparsity ratio settings on mobile in Figure 5. The inference time is measured on the CPU of an off-the-shelf Samsung Galaxy S20 smartphone. Each test takes 50 runs on different inputs with 8 threads on CPU. As different runs do not vary greatly, only the average time is reported. From the results we can see that the obtained sparse model from SparCL can significantly accelerate the inference on both Split-CIFAR-10 and Tiny-ImageNet dataset compared to the model obtained by traditional CL training. For ResNet-18 on Split-CIFAR-10, the model obtained by traditional CL training, which is a dense model, takes 18.53ms for inference. The model provided by SparCL can achieve an inference time of 14.01ms, 8.30ms, and 5.85ms with sparsity ratio of 0.75, 0.90, and 0.95, respectively. The inference latency of the dense ResNet-18 obtained by traditional CL training on Tiny-ImageNet is 39.64 ms. While the sparse models provided by SparCL with sparsity ratio settings as 0.75, 0.90, and 0.95 reach inference speed of 33.06ms, 20.37ms, and 15.49ms, respectively, on Tiny-ImageNet.

## D Comparison with Buffer Selection Methods

In this section, we compare DDR and GSS [3] or Loss-Aware Reservoir Sampling (LARS) [9].

DDR aims at removing training examples for efficiency, while GSS and LARS put the focus on selecting more informative examples that are saved in the buffer. Technically, DDR removes less informative training examples at certain epochs (and thus indirectly affects samples saved in the buffer), while GSS and LARS directly replaces less informative buffered examples in the buffer. Thus, the original GSS and LARS are not directly comparable to DDR. However, we can actually use the example importance criteria used in GSS and LARS to remove less informative training examples as well. We replace the misclassification rate in DDR by the gradient-based (GSS) and loss-based criteria (LARS) objectives and get two variants of our approach, DDR-GSS and DDR-LARS, respectively. For fair comparison, we fix all other parameters used in DDR the same for all methods (sparsity 0.75, remove 30% training data, with TDM only). Since all variants of DDR already remove training examples for efficiency, we mainly focus on their accuracy performance here. The final results on Split-CIFAR10 is shown in the table below:

Table A4: Comparison with Buffer Selection Methods.

| Method | Importance | Accuracy |
|---|---|---|
| DDR | Misclassification | 73.80 |
| DDR-GSS | Gradient | 73.45 |
| DDR-LARS | Loss | 73.67 |

# E  Exploration on Pruning Pattern

In this work, we conduct uniform pruning (i.e., each layer has the same pruning ratio) across different CONV layers as mentioned before in experimental details. The usage of uniform pruning ratio is to match the single-instruction multiple-data (SIMD) [47] architecture of embedded CPU/GPU processors for efficient hardware accelerations.

To observe the pruning pattern, we also experimented with setting an overall pruning ratio as $95\%$ for the entire network, allowing each layer to have different pruning ratios by ranking CWI for the whole model. According to the results, earlier CONV layers tend to have a smaller pruning ratio, which is only around $25\% - 30\%$, while the pruning ratios for the latter CONV layers can reach $99\%$. The results are reasonable, as latter layers are more redundant with a larger amount of parameters. In addition, the weights in earlier layers might be more important for keeping high accuracy, but take a large portion of the computation. Therefore, though slightly improving the accuracy performance to $72.45\%$ compared to the uniform pruning ratio setting, allowing different pruning ratios across different layers yields worse acceleration (drop to $2.2\times$ compared with $3.1\times$ when adopting the uniform pruning ratio) on the hardware. As our purpose is to facilitate the efficiency of the CL-system, we adopt the uniform pruning ratio setting.