# OpenReview forum: "SparCL: Sparse Continual Learning on the Edge"
_NeurIPS.cc/2022/Conference — NeurIPS 2022 Accept_

### Official Review · Reviewer_8efq · 2022-06-26

**Rating:** 6
**Confidence:** 4
**Soundness:** 2 fair
**Presentation:** 3 good
**Contribution:** 3 good

**Summary:**

The paper addresses continual learning in edge devices. It claims that previous work on CL didn't report the actual FLOPs in training which is very important in real life applications on edge devices. The paper includes adaptations of sparse training to CL . It applies three techniques:  1)dynamic weight sparsification based on the magnitude of the weights and magnitude of the gradients on the novel task and on the buffer (it combines it with collapse and expand approach on class boundaries). 2)It reduces the data based on misclassification. 3)It applies less weight updates by not updating parameters with low gradients. Applying these techniques results in significant time and memory savings without loss of accuracy when integrated with rehearsal methods for CL. It even shows slight improvement in forgetting.



**Questions:**

There is no section 6 in the Appendix.

**Limitations:**

Yes

**Strengths And Weaknesses:**

Strength: reduces the resources in CL, while previous work focused on not increasing them. Useful in real-life applications. Novel in terms of focus.
Weakness: The proposed techniques  are straightforward and  do not offer technical novelty.
However, suggesting a straightforward solution in a paper that raises a new problem/challenge (to the best of knowledge) could still be OK.

---

> ### Author Response · Authors · 2022-08-02
> **Response to Reviewer 8efq**
>
> > Q1: The proposed techniques are straightforward and do not offer technical novelty. However, suggesting a straightforward solution in a paper that raises a new problem/challenge (to the best of knowledge) could still be OK
>
> R1:
> We sincerely thank the reviewer for recognizing the new problem and challenge we are trying to solve. However, we would still like to point out the novelty of our work beyond solving the crucial yet under-investigated efficiency problem in CL.
>
> Our proposed method serves as a general framework that introduces multiple levels of sparsity into CL, and greatly improves existing CL methods in efficiency while maintaining accuracy. On the other hand, existing representative sparse training methods do not consider the CL setting;  as a result, they suffer from significant accuracy drop when transfered to the CL setting.
> Moreover, our method is model-agnostic and could improve CL methods of all kinds: Besides rehearsal-based methods explored in the paper, we added additional experiments to demonstrate the effectiveness of our method on non-rehearsal based [1] and representation learning based methods [2] (please see the Appendix F and G of the updated paper). These results demonstrate that SparCL consistently improves different kinds of CL methods under different settings, further indicating the generality of our method.
>
> Besides comprehensive empirical study on benchmark datasets, we evaluate SparCL on a real mobile edge device, demonstrating the practical potential of our method. We believe our work is an important pilot work that encourages future research on CL on-the-edge.
>
> > Q2: There is no section 6 in the Appendix.
>
> R2: Sorry for the confusion. Section 6 is actually the conclusion section, not in the appendix.
>
> Please let us know if all of your concerns have been addressed and we are happy to further discuss and clarify. We look forward to your reply.
>
> **References**
>
>
> [1] Kirkpatrick, James, et al. "Overcoming catastrophic forgetting in neural networks." Proceedings of the national academy of sciences 114.13 (2017): 3521-3526.
>
> [2] Cha, Hyuntak, Jaeho Lee, and Jinwoo Shin. "Co2l: Contrastive continual learning." Proceedings of the IEEE/CVF International Conference on Computer Vision. 2021.

---

> ### Author Response · Authors · 2022-08-07
> **Looking Forward to Your Feedback!**
>
> Dear Reviewer 8efq,
>
> Thank you very much for reviewing our paper and recognizing the contributions of our work. In our posted response earlier, we have clarified the technical novelty of our method. Furthermore, we added additional experiments in the updated paper to demonstrate the generality of our method. We hope that you can find our response convincing. If you have any additional comments, feel free to let us know. We look forward to discussing with you and will try our best to address any further concerns before the discussion deadline.
>
> Thank you very much,
>
> Authors

---

### Official Review · Reviewer_igAw · 2022-07-11

**Rating:** 7
**Confidence:** 4
**Soundness:** 3 good
**Presentation:** 3 good
**Contribution:** 3 good

**Summary:**

The authors discuss whether continual learning can be investigated under the perspective of training efficiency and propose combining network sparsification techniques in a CL setting. As a result, they notice that sparser network are not necessarily worse continual learners and that they can possibly surpass some recent non-sparse approaches. The effectiveness of the proposed SparCL method is investigated by means of ablative studies and a conclusive test on an edge device.

**Questions:**

Please refer to the weakness section above, I am mostly interested in seeing the following point addressed:

- Why is DER++ not among competitors?
- Can any hypothesis be made on whether representation learning approaches CO2L/Dualnet can also benefit from sparsification?
- Where are weights pruned the most in the backbone, is there a pattern?


**Limitations:**

The authors adequately addressed the limitations and potential negative societal impact of their work. It is recommended to do so in the main paper, not in the appendix.

**Strengths And Weaknesses:**

Strenghts
- At the core of this work is a very well-thought intuition: the learning process is noisy/overparameterised and pruning can be beneficial in CL, where we need to cram knowledge in waves into a model and possibly retain generalisation capabilities to facilitate upcoming tasks.
- the presentation of this work is clear, the exposition is easy to follow.

Weaknesses (in decreasing order of importance)
- While the authors claim that they are using a SOTA roster of competitors, I believe that this is not exactly the case. Some recent methods that are missing and could provide an interesting comparison are: LUCIR [a]/BiC [b] (somewhat stronger baselines w.r.t. iCaRL that additionally manage the bias problem), MIR [c]/GSS [d] (which similarly to DDR operate a selection on data which is memorised), CO2L [e]/DualNet [f] (recent works focussing on representation learning, which could or could not benefit from sparsification, as their inner working is rather different from standard replay methods).
- In line with the previous point, the authors take DER as one of their reference baselines. In doing so, however, they are omitting the improved DER++ method proposed in the same paper that appears to be better in all respects w.r.t. DER++. Is there a reason for this exclusion? If not, I take that DER++ should be included instead.
- It would be very interesting to have an additional study on where the sparsification takes place in the network. The authors describe very practical routines to establish which weights to prune. Linked to the recent trend discussing the effect of catastrophic forgetting on different layers of the networks [g], it would be very interesting to have an analysis of at which level the network is pruned the most and whether this changes at all within or between tasks.
- an experimental comparison could be made between DDR and the approach for buffer reduction delineated in [d] (gradient based) or the one described in [h] (loss based)
- there seem to be some problems with Alg.1: the second if-then (if $t=\delta$ k) is missing its end and I cannot understand whether the following $t mod \delta k = 0$ also applies for tasks > 1. If this is the case -- as I understood from the text -- then this is not an else-if.
- I recommend that the authors include figures in this paper in vectorial format so as to facilitate reading.
- as per neurips policies, limitations and societal impact should be in the main paper, not in the appendix
- Typos: line 95 (thus not applicable - verb missing), line 161 (missing escape of space after w.r.t. - write w.r.t.\ in latex), line 282 (contributes -> contribute), line 288 (as much as informative -> as much informative)

[a] Hou, Saihui, et al. "Learning a unified classifier incrementally via rebalancing." Proceedings of the IEEE/CVF Conference on Computer Vision and Pattern Recognition. 2019.
[b] Wu, Yue, et al. "Large scale incremental learning." Proceedings of the IEEE/CVF Conference on Computer Vision and Pattern Recognition. 2019.
[c] Aljundi, Rahaf and Belilovsky, Eugene and Tuytelaars, Tinne and Charlin, Laurent and Caccia, Massimo and Lin, Min and Page-Caccia, Lucas, "Online Continual Learning with Maximal Interfered Retrieval", NEURIPS 2019
[d] Aljundi, Rahaf, et al. "Gradient based sample selection for online continual learning." Advances in neural information processing systems 32 (2019).
[e] Cha, Hyuntak, Jaeho Lee, and Jinwoo Shin. "Co2l: Contrastive continual learning." Proceedings of the IEEE/CVF International Conference on Computer Vision. 2021.
[f] Pham, Quang, Chenghao Liu, and Steven Hoi. "Dualnet: Continual learning, fast and slow." Advances in Neural Information Processing Systems 34 (2021): 16131-16144.
[g] Ramasesh, Vinay V., Ethan Dyer, and Maithra Raghu. "Anatomy of catastrophic forgetting: Hidden representations and task semantics." arXiv preprint arXiv:2007.07400 (2020).
[h] Buzzega, Pietro, et al. "Rethinking experience replay: a bag of tricks for continual learning." 2020 25th International Conference on Pattern Recognition (ICPR). IEEE, 2021.

---

> ### Author Response · Authors · 2022-08-02
> **Response to Reviewer igAw - Part 2**
>
> > Q4: an experimental comparison could be made between DDR and the approach for buffer reduction delineated in [d] (gradient based) or the one described in [h] (loss based)
>
> R4:
> We agree that it is interesting to compare DDR and GSS [d] or Loss-Aware Reservoir Sampling (LARS) [h], however, we would like to kindly point out that the objective of DDR and these methods are different:
> As discussed in our paper (lines 182-185), DDR aims at removing *training examples* for efficiency, while GSS and LARS put the focus on selecting more informative examples that are saved in the buffer.
> Technically, DDR removes less informative *training examples* at certain epochs (and thus indirectly affects samples saved in the buffer), while GSS and LARS directly replaces less informative *buffered examples* in the buffer.
> Thus, the original GSS and LARS are not directly comparable to DDR. However, we can actually use the example importance criteria used in GSS and LARS to remove less informative training examples as well. We replace the misclassification rate in DDR by the gradient-based (GSS) and loss-based criteria (LARS) objectives and get two variants of our approach, DDR-GSS and DDR-LARS, respectively. For fair comparison, we fix all other parameters used in DDR the same for all methods (sparisity 0.75, remove $30$% training data, with TDM only). Since all variants of DDR already remove training examples for efficiency, we mainly focus on their accuracy performance here. The final results on Split-CIFAR10 is shown in the table below:
> | Method    | Importance    |      Accuracy     |
> |-|-|-|
> |   DDR         |      Misclassification   | $73.80$  |
> |   DDR-GSS  |      Gradient    | $73.45$  |
> |   DDR-LARS     |      Loss    | $73.67$  |
>
> We also include these additional results in Appendix H of our updated paper. We can see our basic strategy outperforms other variants, indicating that our DDR strategy is simple and effective. Moreover, the actual difference between the variants are very small and all variants lead to both accuracy and efficiency improvement. This observation further demonstrates DDR is even robust to different example importance criteria. Moreover, DDR actually serves as a general strategy for training data removal in CL and can be further improved by future work on better learning and representing example importance.
>
> > Q5: Fix algorithm.
>
> R5:
> Thanks for the catch. We have fixed these typos by adding ends to both if’s and fixed the else-if to if in our updated version.
>
> > Q6: Fix figure format.
>
> R6:
> Thanks for the kind reminder. During the submission, we tried to upload our paper with all figures in vectorial format. However, Figure 1 was not displaying correctly and left a blank in the vectorial format. So we instead used the .png format for Figure 1. We will troubleshoot this issue and make sure it is in vectorial format in the camera ready version.
>
> > Q7: Limitations and societal impact position.
>
> R7:
> Thanks for the suggestion. When submitting the paper, we referred to the submission FAQs (https://neurips.cc/Conferences/2022/PaperInformation/NeurIPS-FAQ) and found out the following instruction:
>
> *You may include a discussion of these potential negative societal impacts anywhere in the paper (in the intro, in the conclusion, as a stand-alone section, *in the supplemental material if appropriate*, etc.)*
>
> Thus we decided to put these sections  in the Appendix. Nevertheless, we are happy to move them to the main paper after we finalize all other changes in the final version.
>
>
> > Q8: Typos.
>
> R8:
> We have fixed these typos in the updated version.
>
>
> Please let us know if all of your concerns have been addressed and we are happy to further discuss and clarify. We look forward to your reply.
>
> **References**
>
> [1] Wei Niu, Xiaolong Ma, et al.  Patdnn: Achieving real-time dnn execution on mobile devices with pattern-based weight pruning. ASPLOS’ 2020

---

> > ### Comment · Reviewer_igAw · 2022-08-04
> > **Feedback**
> >
> > Thank you for addressing my suggestions, I believe that the new experiments provided further improve the quality of your submission (with particular reference to the ones on CO2L and the localization of pruning). I confirm that I recommend acceptance, I am open to discussing with other reviewers if they want an additional opinion on any point they raised.
> >
> > Also, thanks for pointing out the FAQs about the placement of the limitation section, you may disregard my initial request for moving it.

---

> > > ### Author Response · Authors · 2022-08-05
> > > **Additional Response to Reviewer igAw**
> > >
> > > We sincerely thank the reviewer for recognizing the further improvement of the quality of our work and confirming acceptance! We also appreciate the reviewer’s support of our work by being open to discussing with other reviewers to provide additional opinions.

---

> ### Author Response · Authors · 2022-08-02
> **Response to Reviewer igAw - Part 1**
>
> We sincerely appreciate your careful review and a great summary of our contributions. And thank you very much for the very constructive comments, which are addressed below.
>
> > Q1: Comparison with more recent SOTA, e.g., representation learning method, CO2L.
>
> R1: We thank the reviewer for these great references. To further demonstrate the generality of SparCL, we include the comparison between CO2L and SparC-CO2L at a sparsity ratio of 0.75.
> The results on Split-CIFAR10 are shown in the table below:
> | Method    | Buffer    |      Class-IL         |  FLOPs Train |
> |-|-|-|-|
> |   CO2L         |       $0$    | $58.89$  |  $3.3\times 10^{16}$|
> |    SparCL-CO2L$_{75}$   |       $0$    | $59.43$  |  *$0.6\times 10^{16}$*|
> |   CO2L         |       $200$    | $65.57$  |  $4.4\times 10^{16}$|
> |    SparCL-CO2L$_{75}$   |       $200$    | $66.03$  |  *$0.8\times 10^{16}$*|
> |   CO2L         |       $500$    | $74.26$  |  $4.4\times 10^{16}$|
> |    SparCL-CO2L$_{75}$   |       $500$    | $75.87$  |  *$0.8\times 10^{16}$*|
>
> We also include these additional results in Appendix F of our updated paper. From these results, we can see that SparCL consistently improves CO2L with different buffer sizes, in terms of both accuracy and training FLOPs. The result further indicates the generality of SparCL that it even improved representation learning approaches. We are happy to explore how SparCL would improve more different kinds of CL methods to gain more empirical and theoretical insights as our future work.
>
> > Q2: Why is DER++ not among competitors?
>
> R2: Our apologies for the confusion; the “DER” in our paper actually refers to the stronger “DER++”. We have revised the naming in the updated version.
>
>
> > Q3: Where are weights pruned the most in the backbone, is there a pattern?
>
> R3:
> In this work, we conduct uniform pruning (i.e., each layer has the same pruning ratio) across different CONV layers as mentioned in line 250 in experimental details. The usage of uniform pruning ratio is to match the single-instruction multiple-data (SIMD) [1] architecture of embedded CPU/GPU processors for efficient hardware accelerations.
>
> To observe the pruning pattern, we also experimented with setting an overall pruning ratio as $95$% for the entire network, allowing each layer to have different pruning ratios by ranking CWI for the whole model. We include these additional results in Appendix I of our updated paper. According to the results, earlier CONV layers tend to have a smaller pruning ratio, which is only around $25-30$%, while the pruning ratios for the latter CONV layers can reach $99$%. The results are reasonable, as latter layers are more redundant with a larger amount of parameters. In addition, the weights in earlier layers might be more important for keeping high accuracy, but take a large portion of the computation. Therefore, though slightly improving the accuracy performance to $72.45$% compared to the uniform pruning ratio setting, allowing different pruning ratios across different layers yields worse acceleration (drop to $2.2\times$ compared with $3.1\times$ when adopting the uniform pruning ratio) on the hardware. As our purpose is to facilitate the efficiency of the CL-system, we adopt the uniform pruning ratio setting.

---

### Official Review · Reviewer_g1tN · 2022-07-11

**Rating:** 5
**Confidence:** 5
**Soundness:** 3 good
**Presentation:** 3 good
**Contribution:** 2 fair

**Summary:**

The paper  proposes Sparse Continual Learning (SparCL in which they wanted to do cost-effective continual learning on edge devices by leveraging the  sparsity. Author's also claim that they achieve both training acceleration and accuracy preservation through the synergy of three aspects: weight sparsity, data efficiency, and gradient sparsity. Including TDM process for dynamic data removal (DDR) to remove less informative training data, and dynamic gradient masking (DGM) to sparsify the gradient updates. Authors also claim that the proposed method is uses 23Xless training FLOPs, and, achieving the SOTA accuracy by at most 1.7%.  The authors also tested the method on two datasets split cifar-10, spilt tiny Image-net in Class Incremental and Task Incremental setting.

**Questions:**


Please refer "Strengths And Weaknesses"

**Limitations:**

Please see "Strengths And Weaknesses"

**Strengths And Weaknesses:**



Strengths:

+ Inclusion of Dynamic data removal along with sparsity is good benefit to overall system.
+ Sparsity in gradient and propose dynamic gradient masking
+ Method also considers the  importance of weights w.r.t. data saved in rehearsal buffer.

Con's:
- While reading the paper, I really felt that some sections of the paper are not clearly written specifically Sec 4.1 Task-aware Dynamic Masking.

- I am not really clear/convincing about the authors claim about the memory foot print and training FLOPS. Consider one of the longback papers basic paper on CL using Neural Pruning (https://arxiv.org/abs/1903.04476)(You can choose any other regular CL on pruning paper) and some other CL using sparsity papers which does CL on split CIFAR-10/CIFAR-100 using pruning and they achieve the CL on them without using memory buffer of 500 images. Similarly in the method proposed by authors in which it is also showed that that they need to do extra steps of dynamic Gradient Masking,Dynamic Data Removal which add the overhead too. Clarify this if I am wrong in my understanding on how these additional overheads won't effect the training FLOPS and memory footprint.

- I am also not clear with authors point of view of choosing the "Rehearsal based methods also" for comparing for  "memory foot print and training" as it is obviously clear that rehearsal based methods does need  additional amount of training and memory.

- I am also very curious to know what happens when buffer size of the proposed method is 0. What happens to accuracies as shown in the Table 1 when buffer size increased from 200--->500 the acc increased from 65-->72

- The improvement in the acc w.r.t to SOTA minimal as PackNet shown in table 1 does show packnet, LPF achieves 93.73%,94.50% acc when compared to best SparCL-DER75(95.19±0.34)(500 buffer) and   SparCL-DER75(94.06±0.45)(200 buffer).

- One more limitation I see regarding the proposed approach is towards the scalability as you can see current papers such as ANML(https://ecai2020.eu/papers/939_paper.pdf), of CL does show CL >100's of tasks. But how would given proposed scale comparing the overhead of buffer, memory masks while trying to learn huge no of tasks.



Overall the paper is generally well written and easy to follow apart from few sections., and the experiments are thorough and well-executed but limited to fewer datasets, lesser tasks. and minimal improvement on SOTA There are extensive ablation experiments demonstrating some of the key components the proposed approach.

---

> ### Author Response · Authors · 2022-08-02
> **Response to Reviewer g1tN - Part 4**
>
> > Q6: One more limitation I see regarding the proposed approach is towards the scalability as you can see current papers such as ANML(https://ecai2020.eu/papers/939_paper.pdf), of CL does show CL >100's of tasks. But how would given proposed scale comparing the overhead of buffer, memory masks while trying to learn huge no of tasks.
>
> R6:
> Thanks for pointing us to this work. It is very important to note the following aspect of the experimental setting used in ANML: although they claim to use a sequence of 600 tasks by splitting the Omniglot dataset, there is **actually only one class per task** (see Figure 2 and 3 in [10]). As a result, the total number of classes used is 600 classes. In our setting, we use Split Tiny-ImageNet, a standard, challenging CL benchmark widely used in the community [4, 6, 11], which has 200 classes split into 10 tasks.  In this sense, the benchmark we use and the dataset in ANML are actually quite comparable in terms of total number of classes to learn.
>
> Moreover, we would like to emphasize that the problem we study is of fundamental importance, and would of course be crucial to all and any attempts at scalability!  The ultimate goal of SparCL is to greatly enhance efficiency of existing CL methods, while keeping accuracy. Thus, the effectiveness of SparCL is actually independent of the setting (long or short task sequences) and the exact CL method (with or without buffer). The fact that we propose a more efficient training framework would help, not hinder, any attempts at scaling the number of tasks or classes.
>
> As for the reviewer’s specific concern on the overhead when there are huge number of tasks, we would also like to further clarify with regard to the buffer and the masks:
> - As clarified in R2, the buffer is *not* a specific overhead of our method, but the rehearsal-based methods that SparCL combined with. Since we already showed in R4 that our SparCL actually works well with methods with buffer size = 0, the overhead of the buffer really depends on the specific methods and how they manage the buffer with long sequences, instead of how SparCL handles it. .
> - As for the cost of masks, please note that we only have 2 masks (TDM and DDR), and the number of masks does *not* scale with the number of tasks, since we are dynamically adjusting the same masks when new tasks come, instead of adding new masks like prior work did [1, 2]. Moreover, the masks are typically saved in a low-bit (consider it only contains 0 and 1) and sparse manner (see Appendix E.1.1), which lead to minimal memory overhead.
> However, when the number of tasks scales up, SparCL could apply a mask with a lower sparsity ratio to provide higher network capacity. We treat this scalability problem in CL and how to better adapt SparCL for greater scalability as interesting future work.
>
>
> Please let us know if all of your concerns have been addressed and we are happy to further discuss and clarify. We look forward to your reply.
>
> **References**
>
> [1] Mallya, Arun, and Svetlana Lazebnik. "Packnet: Adding multiple tasks to a single network by iterative pruning." Proceedings of the IEEE conference on Computer Vision and Pattern Recognition. 2018.
>
> [2] Wang, Zifeng, et al. "Learn-prune-share for lifelong learning." 2020 IEEE International Conference on Data Mining (ICDM). IEEE, 2020.
>
> [3] Chaudhry, Arslan, et al. "On tiny episodic memories in continual learning." arXiv preprint arXiv:1902.10486 (2019).
>
> [4] Buzzega, Pietro, et al. "Dark experience for general continual learning: a strong, simple baseline." Advances in neural information processing systems 33 (2020): 15920-15930.
>
> [5] Kirkpatrick, James, et al. "Overcoming catastrophic forgetting in neural networks." Proceedings of the national academy of sciences 114.13 (2017): 3521-3526.
>
> [6] Cha, Hyuntak, Jaeho Lee, and Jinwoo Shin. "Co2l: Contrastive continual learning." Proceedings of the IEEE/CVF International Conference on Computer Vision. 2021.
>
> [7] Mai, Zheda, et al. "Online continual learning in image classification: An empirical survey." Neurocomputing 469 (2022): 28-51.
>
> [8] Van de Ven, Gido M., and Andreas S. Tolias. "Three scenarios for continual learning." arXiv preprint arXiv:1904.07734 (2019).
>
> [9] Wu, Yue, et al. "Large scale incremental learning." Proceedings of the IEEE/CVF Conference on Computer Vision and Pattern Recognition. 2019.
>
> [10] Beaulieu, Shawn, et al. "Learning to Continually Learn." ECAI. 2020.
>
> [11] De Lange, Matthias, et al. "A continual learning survey: Defying forgetting in classification tasks." IEEE transactions on pattern analysis and machine intelligence 44.7 (2021): 3366-3385.

---

> ### Author Response · Authors · 2022-08-02
> **Response to Reviewer g1tN - Part 3**
>
> > Q4: I am also very curious to know what happens when buffer size of the proposed method is 0. What happens to accuracies as shown in the Table 1 when buffer size increased from 200--->500 the acc increased from 65-->72
>
> R4:
> Please note that for rehearsal-based methods, it is well-known that increasing buffer size leads to accuracy improvement (see, e.g., [3, 4, 7]).  Also, we would like to clarify that in Table 1, our method is combined with ER [3] and DER [4]. They are rehearsal-based methods that are not compatible with buffer size = 0. Nevertheless, our method, SparCL, is compatible with buffer size = 0 simply by removing buffer-related terms in equation (1) and (3) in the main paper. To demonstrate the generality of our method and see how it performs with buffer size = 0, we combine SparCL with another recent SOTA method, CO2L [6], that is indeed compatible with buffer size = 0. The results are shown in the following table:
>
> | Method    | Buffer    |      Class-IL         |  FLOPs Train | Mem-Footprint |
> |-|-|-|-|-|
> |   CO2L         |       $0$    | $58.89$  |  $3.3\times 10^{16}$| 213MB |
> |    SparCL-CO2L$_{75}$   |       $0$    | $59.43$  |  *$0.6\times 10^{16}$*|110MB|
> |   CO2L         |       $200$    | $65.57$  |  $4.4\times 10^{16}$|293MB|
> |    SparCL-CO2L$_{75}$   |       $200$    | $66.03$  |  *$0.8\times 10^{16}$*|186MB|
> |   CO2L         |       $500$    | $74.26$  |  $4.4\times 10^{16}$|293MB|
> |    SparCL-CO2L$_{75}$   |       $500$    | $75.87$  |  *$0.8\times 10^{16}$*|186MB|
>
> We also include these additional results in Appendix F of our updated paper. The results demonstrate that our SparCL can constantly improve the accuracy and efficiency regardless of the buffer size. We also want to point out that introducing the buffer (with buffer size > 0) indeed adds overhead in both FLOPs and memory footprint, because of the additional loss terms that are related to buffered examples (see [4, 6] for more details). However, the calculation of training FLOPs and memory footprint is independent of the buffer size, when buffer size > 0. This is because increasing the buffer size does not affect total iterations of the training process and batch size per iteration. Please refer to Appendix D.1, D.2 for more details about the calculation of FLOPs and memory footprint.
>
>
> > Q5: The improvement in the acc w.r.t to SOTA minimal as PackNet shown in table 1 does show packnet, LPF achieves 93.73%,94.50% acc when compared to best SparCL-DER75(95.19±0.34)(500 buffer) and SparCL-DER75(94.06±0.45)(200 buffer).
>
> R5:
> We would also like to kindly clarify that PackNet [1] and LPS [2] are specialized and limited to the Task-IL setting (where task identity is given at test time), which is much easier than the Class-IL setting (where task identity is *not* given at test time); please see [8] for a detailed comparison of these two settings. These two methods are not compatible with the Class-IL setting. On the contrary, rehearsal-based methods maintain a buffer to address the harder Class-IL setting. In our paper, we only show the Task-IL accuracy for completeness and use Class-IL accuracy as our main accuracy metric, following more recent prior work [4, 6, 9]. In Table 1, SparCL-DER$_{75}$ outperforms all methods in terms of Class-IL accuracy, including PackNet and LPS since they are not compatible with this setting.
>
> Moreover, we would like to clarify the focus of our paper and Table 1: SparCL is not trying to outperform all SOTA methods in terms of accuracy (although SparCL achieves this goal by combining with DER at $0.75$ sparsity ratio). Instead, SparCL serves as a general framework to improve the efficiency of existing CL methods while maintaining accuracy. Thus, accuracy improvement should not be treated as a hard requirement of our method. Please see the efficiency gap reflected in training FLOPs of our methods and other methods: SparCL-DER is $2-8\times$ more efficient than PackNet and LPS with different sparsity ratios, and also how SparCL maintains and even improves accuracy in the more challenging Class-IL setting.

---

> ### Author Response · Authors · 2022-08-02
> **Response to Reviewer g1tN - Part 2**
>
> > Q1: Sec 4.1 Task-aware Dynamic Masking not clear.
>
> R1:
> We have carefully proofread and revised the corresponding section by adding more explanations and fixing typos in Algorithm 1. Please let us know if there are any more detailed concerns or suggestions.
>
> > Q2: Claim about the memory foot print and training FLOPS. Memory buffer overhead and extra steps overhead. Clarify this if I am wrong in my understanding on how these additional overheads won't effect the training FLOPS and memory footprint.
>
> R2:
> We would like to emphasize that we have already taken into account all overheads when computing training FLOPs in Table 1-3; similarly, we have taken into account all overheads when computing the. memory footprint in Table 3 (please refer to Appendices D.1, D.2 for more details about the calculation of FLOPs and memory footprint).  Please note that SparseCL still  outperforms all other methods in terms of both FLOPs and memory footprint. In Table 1, our method (SparCL-DER$_{95}$) achieves training FLOPs that is at least 7$\times$ less than PackNet [1] and LPS [2], which are also CL on pruning algorithms without a buffer or any of the extra steps introduced by our method.
>
>
>
> More specifically:
>
> 1) For the first concern on the memory buffer, we would like to restate that SparCL itself does not require any form of buffer. Instead, SparCL serves as a general framework that is compatible with existing CL methods, with or without a buffer, to *improve* their efficiency while maintaining accuracy. Therefore, **the memory buffer is not a specific overhead of our method; it is simply a feature of the rehearsal-based methods that we combined with SparCL**. In an additional experiment (see also R4), we combine SparCL with a CL method without any buffer, and SparCL consistently yields improvements in both accuracy and efficiency.
>
> 2) For the second concern on “extra steps” of TDM, DDR and DGM, all the three techniques are updated every $\delta k$ epochs as mentioned in lines 166 and 188. The total FLOPs for TDM, DDR, and DGM on split CIFAR-10 during the training process is approximately $4.5\times 10^{9}$, which is **negligible** compared to the entire training FLOPs (total training FLOPs $ >10^{15}$, therefore less than $0.0001$%)  as illustrated in Table 1.  For the memory overhead, DDR only requires the indices (stored in int8 format and approximately 3KB) for the easier examples while TDM and DGM only rely on the existing parameters and do not incur additional memory cost. We also include these results in Appendix D.1 and D.2 of our updated paper.
>
>
> > Q3: I am also not clear with authors point of view of choosing the "Rehearsal based methods also" for comparing for "memory foot print and training" as it is obviously clear that rehearsal based methods does need additional amount of training and memory.
>
> R3:
> Again, we would like to emphasize the focus of our paper is **not** directly comparing our method against rehearsal-based, or any other specific CL methods. Instead, we focus on designing a sparse training framework to **improve** existing CL methods in terms of efficiency without compromising accuracy. We mainly use rehearsal-based methods as our backbone because of their effectiveness in the challenging and practical Class-IL setting (see R5 for detailed explanation). No matter how much memory footprint or training FLOPs the base methods (such as DER [3] or ER [4]) have, we do take them into account for both the base method, and the corresponding version combined with SparCL (e.g. DER v.s. SparCL-DER). Therefore, we are not cherry-picking rehearsal methods because they have additional training and memory cost and thus are easy to outperform. From the experiments, our method reduced both training FLOPs (Table 1) and memory footprint (Table 3) of the base methods, even with an accuracy improvement. Notably, all variants of SparCL-DER in Table 1 outperform other non-rehearsal based methods (LPS [2], EWC [5], etc.) that already have less training FLOPs than DER, by a large margin in terms of training FLOPs.
> To further demonstrate the generality of our method beyond improving rehearsal-based methods, we also conduct additional experiments by combining our framework with CO2L [6] (without buffer); see R4 for more detailed explanation.

---

> ### Author Response · Authors · 2022-08-02
> **Response to Reviewer g1tN - Part 1**
>
> We thank the reviewer for recognizing the strengths of our paper and their valuable feedback. Before we go into details about per-question response, we would like to clarify several crucial points
> - Our method, SparCL, is a general framework for enhancing the efficiency for all kinds of CL methods. Thus, **the effectiveness of SparCL does not depend on the existence or size of the rehearsal buffer**. Put differently, the overhead that the buffer introduces is not because of SparCL (see also R2, R3 below). Nevertheless, to address the reviewer’s concerns, we  include additional experiments to show the effectiveness of SparCL over CL methods without a buffer (R4).
> - The goal of SparCL is **enhancing efficiency while maintaining accuracy, instead of merely pushing the absolute performance in accuracy higher**. As noted in our response to Reviewer BMhS, accomplishing this is neither trivial nor straightforward with existing (non-CL) sparsification methods.
>
> We respond to each of the reviewer’s questions point-by-point below:

---

> ### Author Response · Authors · 2022-08-07
> **Looking Forward to Your Feedback!**
>
> Dear Reviewer g1tN,
>
> Thank you very much for reviewing our paper and leaving valuable comments. In our posted response (Part 1 - 4) earlier, we have conducted new experiments and added additional clarifications to address your concerns/questions about our original submission. We hope that you can find our response convincing. If you have any additional comments, feel free to let us know. We look forward to discussing with you and will try our best to address any further concerns before the discussion deadline.
>
> Thank you very much,
>
> Authors

---

> ### Author Response · Authors · 2022-08-09
> **Final Reminder for Discussion**
>
> Dear Reviewer g1tN,
>
> Thank you very much for spending time reviewing our paper. Since the discussion will end very soon, we sincerely hope that you have found time to check our detailed response to your previous questions/comments. If you have any further questions, please feel free to let us know. We will try our best to reply to you before the discussion deadline.
>
> Thank you very much,
>
> Authors

---

### Official Review · Reviewer_BMhS · 2022-07-12

**Rating:** 4
**Confidence:** 3
**Soundness:** 2 fair
**Presentation:** 3 good
**Contribution:** 2 fair

**Summary:**

This paper introduces sparse continual learning, a continual learning framework that overcomes catastrophic forgetting of deep networks in the learning of a task stream and accelerates the training and inference.
The efficiency and effectiveness are achieved through three key components that encourage sparse network weight connection, replay buffer selection, and sparse gradient truncation.
The effectiveness is validated on both public benchmarks on real-world edge deivce.

**Questions:**

The proposed method relies on the usage of a reply buffer. As also discussed by the authors in Appendix Section A, the proposed method can potentially be applied to more advanced methods that do not rely on any previous data. Showing the proposed method to be more 'model agnostic might help improve the paper.



**Limitations:**

The authors discussed the limitations in Appendix Section A

**Strengths And Weaknesses:**

Strengths:

- The presentation of the paper is clear and easy to follow.

- The results on mobile devices are a plus


Weakness:

- I see sparsity as a universal and generic way of improving deep model efficiency. Therefore it is not surprising to see introducing sparsity to CL can improve efficiency.

- Among the three techniques proposed by this paper, only the weight sparsity part seems to have a direct impact on overcoming catastrophic forgetting, which is the key to improving continual learning.

- In Table 1, the latest method included in the comparison is from 2020. The quantitative comparison are clearly insufficient to support the claim that the proposed framework 'further improves the SOTA accuracy by at most 1.7%.'

- Overall, this paper delivers a useful message that sparsity can improve the efficiency of CL, and even slightly improve the accuracy. However, the novelty and soundness of the paper do not meet the standard of NeurIPS.

---

> ### Author Response · Authors · 2022-08-02
> **Response to Reviewer BMhS - Part 2**
>
> > Q4: Overall, this paper delivers a useful message that sparsity can improve the efficiency of CL, and even slightly improve the accuracy. However, the novelty and soundness of the paper do not meet the standard of NeurIPS.
>
> R4:
> Our work is novel in the following aspects:
> - We investigate learning efficiency in CL. We are the very first to do so: this is an under-investigated, yet important aspect in CL. It is also crucial for the real-world application and deployment of CL methods in resource-limited platforms.
> - Our proposed method serves as a general framework that introduces multiple levels of sparsity into CL, and greatly improves existing CL methods in efficiency while maintaining accuracy. On the other hand, existing representative sparse training methods cannot be readily used in the CL setting, resulting in a  significant accuracy drop. This is a finding not previously known, that we both first report here, and directly address.
> We provide the following supporting evidence for the soundness of our approach. We believe these to be quite strong, well within experimental evidence provided by usual NeurIPS papers; please let us know if there are any other specific soundness concerns: :
> - We have conducted comprehensive comparison experiments as well as ablation studies to demonstrate the effectiveness of our method. Additional experiment results with the more recent SOTA method CO2L [8] (presented in R3), and methods without buffers [6, 8] (presented in R5) further strengthen our argument in favor of SparCL.
> - As recognized by the reviewer, we evaluate SparCL on a real mobile edge device, demonstrating the practical potential of our method. The techniques and implementation details can also be found in section 5.4 and appendix E.
>
>
> > Q5: The proposed method relies on the usage of a reply buffer. As also discussed by the authors in Appendix Section A, the proposed method can potentially be applied to more advanced methods that do not rely on any previous data. Showing the proposed method to be more 'model agnostic might help improve the paper.
>
> R5:
> Our proposed method, SparCL, as a general framework, can be easily adapted to scenarios without a buffer by removing buffer-related terms in Equation (1) and (3). For completeness, we show the effectiveness of our method upon EWC [6], one of the most well-known rehearsal-free CL methods, and a more advanced CL method, CO2L [8], without a buffer.
> The results on Split-CIFAR10 with 0.75 sparsity ratio are shown in the table below:
> | Method    | Buffer    |      Class-IL         |  FLOPs Train |
> |-|-|-|-|
> |   EWC         |       $0$    | $19.49$  |  $8.3\times 10^{15}$|
> |    SparCL-EWC$_{75}$   |       $0$    | $20.78$  |  *$1.6\times 10^{15}$*|
> |   CO2L         |       $0$    | $58.89$  |  $3.3\times 10^{16}$|
> |    SparCL-CO2L$_{75}$   |       $0$    | $59.43$  |  *$0.6\times 10^{16}$*|
>
> We also include these additional results in Appendix G of our updated paper. SparCL consistently improves both EWC and CO2L, in terms of both accuracy and efficiency. The results also show the model-agnostic potential of SparCL to improve upon all kinds of methods, including both rehearsal and non-rehearsal methods.
>
>
> Please let us know if all of your concerns have been addressed and we are happy to further discuss and clarify. We look forward to your reply.
>
> **References**
>
> [1] Evci, Utku, et al. "Rigging the lottery: Making all tickets winners." International Conference on Machine Learning. PMLR, 2020.
>
> [2] Lee, Namhoon, Thalaiyasingam Ajanthan, and Philip HS Torr. "Snip: Single-shot network pruning based on connection sensitivity." arXiv preprint arXiv:1810.02340 (2018).
>
> [3] Aljundi, Rahaf, et al. "Gradient based sample selection for online continual learning." Advances in neural information processing systems 32 (2019).
>
> [4] Buzzega, Pietro, et al. "Rethinking experience replay: a bag of tricks for continual learning." 2020 25th International Conference on Pattern Recognition (ICPR). IEEE, 2021.
>
> [5] Mai, Zheda, et al. "Online continual learning in image classification: An empirical survey." Neurocomputing 469 (2022): 28-51.
>
> [6] Kirkpatrick, James, et al. "Overcoming catastrophic forgetting in neural networks." Proceedings of the national academy of sciences 114.13 (2017): 3521-3526.
>
> [7] Zenke, Friedemann, Ben Poole, and Surya Ganguli. "Continual learning through synaptic intelligence." International Conference on Machine Learning. PMLR, 2017.
>
> [8] Cha, Hyuntak, Jaeho Lee, and Jinwoo Shin. "Co2l: Contrastive continual learning." Proceedings of the IEEE/CVF International Conference on Computer Vision. 2021.
>
> [9] Mostafa, Hesham, and Xin Wang. "Parameter efficient training of deep convolutional neural networks by dynamic sparse reparameterization." International Conference on Machine Learning. PMLR, 2019.

---

> ### Author Response · Authors · 2022-08-02
> **Response to Reviewer BMhS - Part 1**
>
> We thank the reviewer for recognizing the strengths of our paper and the feedback. We are happy to address weaknesses and questions raised below:
>
> > Q1: I see sparsity as a universal and generic way of improving deep model efficiency. Therefore it is not surprising to see introducing sparsity to CL can improve efficiency.
>
> R1:
> We agree that sparsity is a generic way of improving deep model efficiency. However, we would like to emphasize that **there is no guarantee that introducing sparsity to training can maintain model accuracy**.   Naive ways of introducing sparsity can lead to a significant decrease in accuracy, and there is a large body of research on how to do this in the standard learning (non-CL) setting (see, eg., [1,2], but also [9]). Most importantly, our method is not a trivial combination of such sparsity techniques and CL. Applying such methods directly comes with a severe accuracy degradation. For example, applying [1,2] directly to the CL setting leads to at least a  $5$% lower accuracy than our SparCL, with $1.5\times$ more training FLOPs (see Table 2). Notably, SparCL maintains accuracy even at $0.95$ sparsity ratio, i.e. $1/20$ of the original model size while none of the “simple” combinations of sparse training and CL methods achieve similar performance. Moreover, all components of our methods are well-motivated and specially designed for the CL setting; please also refer to our response  R2 to see how we design all three components to both enhance efficiency and overcome forgetting in CL.
>
>
> > Q2: Among the three techniques proposed by this paper, only the weight sparsity part seems to have a direct impact on overcoming catastrophic forgetting, which is the key to improving continual learning.
>
> R2:
> We would like to kindly argue that all three components of SparCL not only enhance training efficiency, but are all well-motivated for improving continual learning by overcoming catastrophic forgetting:
> - Dynamic data removal (DDR) maintains more informative examples and addresses the data imbalance issue between the past and current data; please see lines 194-200 in the paper for a detailed discussion. According to [3, 4], selecting informative examples is crucial to alleviate forgetting. According to [5], addressing the data imbalance issue between tasks is also important for overcoming forgetting.
> - Dynamic gradient masking (DGM) promotes the preservation of past knowledge by preventing a fraction of weights from updating (lines 215-216). Our strategy actually shares motivation with regularization-based methods like [6, 7] to address forgetting.
> - Most importantly, the ablation study in Table 3 (and appendix D.4) demonstrates all components contribute to *both* efficiency and accuracy, which shows their direct impact on improving CL.
>
>
> > Q3: In Table 1, the latest method included in the comparison is from 2020. The quantitative comparisons are clearly insufficient to support the claim that the proposed framework 'further improves the SOTA accuracy by at most 1.7%.'
>
> R3:
> We thank the reviewer for suggesting that there are more recent CL methods. To strengthen our claim, we include the comparison with one of the more recent SOTA CL methods, CO2L [8], suggested by Reviewer igAw. We set the sparsity ratio of SparCL as 0.75. The results on Split-CIFAR10 are shown in the table below:
> | Method    | Buffer    |      Class-IL         |  FLOPs Train |
> |-|-|-|-|
> |   CO2L         |       $200$    | $65.57$  |  $4.4\times 10^{16}$|
> |    SparCL-CO2L$_{75}$   |       $200$    | $66.03$  |  *$0.8\times 10^{16}$*|
> |   CO2L         |       $500$    | $74.26$  |  $4.4\times 10^{16}$|
> |    SparCL-CO2L$_{75}$   |       $500$    | $75.87$  |  *$0.8\times 10^{16}$*|
>
> We have included these additional results in Appendix F of our updated paper. From these results, we can see that SparCL consistently improves upon CO2L, in terms of both accuracy and efficiency measured in training FLOPs. This further strengthens our claim made in the paper, indicating the generality of our SparCL framework for efficient CL without compromising accuracy.
>
> Moreover, we would like to emphasize that our method serves as a *general framework* which enables existing CL methods to become more efficient, without sacrificing accuracy. In that sense, our experiments aim to demonstrate this improvement in efficiency over a broad array of methods (at no accuracy loss). In fact, the observed improvement in accuracy is a surprising property; it further strengthens our argument that efficiency via sparsification should indeed be pursued.

---

> ### Author Response · Authors · 2022-08-07
> **Looking Forward to Your Feedback!**
>
> Dear Reviewer BMhS,
>
> Thank you very much for reviewing our paper and leaving valuable comments. In our posted response (Part 1 and 2) earlier, we have conducted new experiments and added additional clarifications to address your concerns/questions about our original submission. We hope that you can find our response convincing. If you have any additional comments, feel free to let us know. We look forward to discussing with you and will try our best to address any further concerns before the discussion deadline.
>
> Thank you very much,
>
> Authors

---

> ### Author Response · Authors · 2022-08-09
> **Final Reminder for Discussion**
>
> Dear Reviewer BMhS,
>
> Thank you very much for spending time reviewing our paper. Since the discussion will end very soon, we sincerely hope that you have found time to check our detailed response to your previous questions/comments. If you have any further questions, please feel free to let us know. We will try our best to reply to you before the discussion deadline.
>
> Thank you very much,
>
> Authors

---

### Meta-Review · Area_Chair_C5NU · 2022-08-27

**Recommendation:** Accept
**Confidence:** Certain

**Metareview:**

This paper introduces a new continual learning scheme whose efficiency and effectiveness are achieved through three key components that encourage sparse network weight connection, replay buffer selection, and sparse gradient truncation. After the author-review discussion phase, a majority of reviewer suggest acceptance. Only one negative reviewer did not respond to the authors' rebuttal, but AC thinks that it is convincing enough to resolve her/his concerns. AC thinks that investigating sparse networks for continual learning is novel, and demonstrating it under edge-device level is a big plus. Overall, AC is happy to recommend acceptance. AC strongly recommend the authors to incorporate all additional results and discussion-with-reviewers into the final draft.

**Award:**

Yes

---

### Decision · Program_Chairs · 2022-09-14

Accept